# Flatworm-specific transcriptional regulators promote the specification of tegumental progenitors in *Schistosoma mansoni*

George R Wendt[1], Julie NR Collins[1], Jimin Pei[2,3], Mark S Pearson[4], Hayley M Bennett[5], Alex Loukas[4], Matthew Berriman[5], Nick V Grishin[2,3], James J Collins III[1]*

[1]Department of Pharmacology, UT Southwestern Medical Center, Dallas, Texas; [2]Department of Biophysics, UT Southwestern Medical Center, Dallas, Texas; [3]Howard Hughes Medical Institute, UT Southwestern Medical Center, Dallas, Texas; [4]Center for Biodiscovery and Molecular Development of Therapeutics, Australian Institute of Tropical Health and Medicine, James Cook University, Cairns, Australia; [5]Wellcome Sanger Institute, Wellcome Genome Campus, Hinxton, United Kingdom

**Abstract** Schistosomes infect more than 200 million people. These parasitic flatworms rely on a syncytial outer coat called the tegument to survive within the vasculature of their host. Although the tegument is pivotal for their survival, little is known about maintenance of this tissue during the decades schistosomes survive in the bloodstream. Here, we demonstrate that the tegument relies on stem cells (neoblasts) to specify fusogenic progenitors that replace tegumental cells lost to turnover. Molecular characterization of neoblasts and tegumental progenitors led to the discovery of two flatworm-specific zinc finger proteins that are essential for tegumental cell specification. These proteins are homologous to a protein essential for neoblast-driven epidermal maintenance in free-living flatworms. Therefore, we speculate that related parasites (i.e., tapeworms and flukes) employ similar strategies to control tegumental maintenance. Since parasitic flatworms infect every vertebrate species, understanding neoblast-driven tegumental maintenance could identify broad-spectrum therapeutics to fight diseases caused by these parasites.

DOI: https://doi.org/10.7554/eLife.33221.001

*For correspondence: JamesJ.Collins@UTSouthwestern.edu

## Introduction

Schistosomes cause significant morbidity and mortality in some 200 million people in the developing world (*Hotez and Fenwick, 2009*). An astounding feature of these parasites is their ability to survive for decades in the vasculature of their human hosts. Indeed, the literature is rife with cases of patients harboring reproductively active schistosomes 20–30 years after leaving endemic regions (*Harris et al., 1984*; *Hornstein et al., 1990*; *Payet et al., 2006*). How these parasites flourish for years in what has been described as 'the most hostile environment imaginable' (*McLaren, 1980*) remains an open question. A skin-like tissue known as the tegument is thought to be key for the schistosome's long-term survival inside the vasculature of the host. The tegument is a continuous syncytium that covers the worm's entire outer surface. While the tegument itself lacks many basic cellular components (i.e., ribosomes, nuclei, endoplasmic reticulum), this tissue is connected via cytoplasmic projections to thousands of individual cell bodies that lay beneath the parasite's muscle layers. These tegumental cell bodies (called 'cytons' in the classic literature) are nucleated and

**eLife digest** Schistosomiasis is a devastating disease that infects more than 200 million people and kills 200 thousand people every year. The disease is caused by parasitic flatworms known as schistosomes. These worms can live in the bloodstream for decades, even if the host has a healthy immune system. This ability to evade the immune system is thought to be partly due to the worm's special 'skin', a tissue referred to as the tegument that all parasitic flatworms have. The tegument is a massive cell that covers the entire surface of the worm, and is thought to be an adaptation that enabled flatworms to become parasites.

Despite the important role that the tegument appears to play in the biology of parasitic flatworms, very little is actually known about how this tissue is made and maintained. The tegument likely experiences a great deal of damage because it serves as the interface between the parasite and the host. Can the parasite repair the tissue as it becomes damaged? If the parasite relies upon this renewal, then preventing schistosomes from repairing their tegument could be a new way to treat schistosomiasis.

Wendt et al. developed a new technique to fluorescently label a schistosome's tegument. This revealed that the parasite does continuously repair and replace its tegument. To better understand this process, Wendt et al. identified genes that were active in the cells responsible for making the tegument. Two of these genes appear to regulate tegument production, and these genes can be found in both parasitic and non-parasitic flatworms.

Further studies of these genes could shed light specifically onto how parasitism arose in flatworms. In addition, a better understanding of how the tegument develops and functions could identify new drug targets that could be used against the many diseases caused by parasitic flatworms.

DOI: https://doi.org/10.7554/eLife.33221.002

provide a continuous stream of proteins and secreted material to support tegumental function (*Wilson and Barnes, 1974*; *McLaren, 1980*). Scientists have long thought that the uninterrupted architecture of the tegument and the unique molecular composition of the tegmental surface are key for evasion of host defenses and parasite survival (*McLaren, 1980*; *Skelly and Alan Wilson, 2006*). Despite these essential functions, little is known on the cellular and molecular level about the development and long-term maintenance of the tegument inside the parasite's definitive host.

Schistosomes are members of the Neodermata (*Ehlers, 1985*; *Littlewood and Bray, 2001*; *Laumer et al., 2015*), a large clade of parasitic Platyhelminthes that includes some of nature's most notorious pathogens, including both tapeworms and some 20,000 species of flukes. Aside from being parasites, all members of the Neodermata are united by the fact that they possess a tegument similar to that of the schistosome. As in schistosomes, the importance of this tegument in the biology of these parasites cannot be overstated. The tegument forms a protective barrier that guards these parasites, not only against the host immune system, but also from the physical extremes they encounter living in the digestive system, blood, or internal organs of their host. The tegument also serves as a conduit for the worms to acquire nutrients (*Halton, 1997*). Indeed, during the course of evolution, tapeworms have lost their gut in favor of utilizing the tegument as their primary means of nutrient acquisition. Moreover, the tegument is rapidly remodeled on a cellular and molecular level when these parasites transition between intermediate and definitive hosts (*Hockley and McLaren, 1973*; *Tyler and Tyler, 1997*; *Tyler and Hooge, 2004*), suggesting that this tissue may also have been pivotal in allowing the complex multi-host lifecycles that are essential for the propagation of these obligate parasites. Given the benefits that the tegument affords these parasites, and its absence in free-living members of the phylum, it is widely credited as the key innovation leading to the evolution of parasitism in the Platyhelminthes (*Tyler and Tyler, 1997*; *Tyler and Hooge, 2004*; *Littlewood, 2006*; *Laumer et al., 2015*). Thus, a deeper understanding of the molecular regulation of tegument development could provide important insights into flatworm evolution and suggest targets for the development of novel anthelmintics.

Upon invasion of their definitive host, the schistosome tegument is rapidly remodeled in a process that appears to be fueled by the fusion of mesenchymal cells to the outer tegument

(*Hockley and McLaren, 1973*; *Skelly and Shoemaker, 2001*). After this fusion takes place, however, little is known about how the cellular composition of the tegument changes during parasite maturation or during the decades that these parasites can potentially live in the vasculature. One important, but virtually unexplored, question is whether the tegument is subject to physiological cell replenishment or turnover. Since the tegument is a syncytium, it is possible that aberrant function (or death) of a small fraction of cells could be compensated for by the remaining cells in the tissue. While this possibility has not been ruled out, recent studies have hinted at a role for stem cells (called neoblasts [*Collins et al., 2013*]) in contributing to the rejuvenation and maintenance of the schistosome tegument (*Collins et al., 2016*). Indeed, the primary differentiation output of neoblasts appears to be a group of short-lived cells that express an mRNA encoding TSP-2, a promising anti-schistosome vaccine candidate that is present at high-levels in the tegument (*Tran et al., 2006*; *Pearson et al., 2012*) and on tegument-derived extracellular vesicles (*Sotillo et al., 2016*). In addition to expressing *tsp-2* mRNA, these neoblast progeny cells express a collection of known tegument-specific factors, suggesting that neoblasts are important in some capacity for contributing to the maintenance of the tegument (*Collins et al., 2016*). However, due to a lack of tools for visualizing both the outer tegument and its attached cell bodies, the relationship between *tsp-2*[+] neoblast progeny and the tegument remains uncharacterized.

Here, we describe a novel methodology to fluorescently label the schistosome tegument and demonstrate that tegumental cells are renewed continuously by a population of *tsp-2*[+] progenitor cells that fuse with the tegument. To define how this process is regulated on a molecular level, we characterized the transcriptomes of both neoblasts and tegumental progenitors using fluorescence-activated cell sorting (FACS). Using these transcriptomes as a guide, we conducted an RNAi screen to discover molecular regulators of tegument differentiation, and identify a pair of flatworm-specific zinc finger proteins, called ZFP-1 and ZFP-1–1, that are essential for the specification of new tegumental cells. Since these zinc finger proteins are flatworm-specific, and a homolog of these proteins is known to be essential for a very similar epidermal biogenesis program in free-living flatworms, we speculate that these genes are likely to be key for tegument development across the Neodermata. Our data demonstrate a formal role for neoblasts in tegumental maintenance and provide the first molecular insights into how tegumental fates are specified.

## Results

### The schistosome tegument and associated cell bodies can be labeled specifically with fluorescently conjugated dextran

A prerequisite for studying the development of the tegument is the ability to visualize both the outer tegument and its associated cell bodies microscopically (*Figure 1A*). However, this presently can only be accomplished by transmission electron microscopy (*McLaren, 1980*), which is not compatible with methodologies to visualize the expression of molecular markers. Therefore, we explored a variety of live cell dyes and delivery techniques to identify an approach to specifically label the schistosome tegument fluorescently (*Figure 1A*). We found that soaking live parasites in a hypotonic solution of 10 kDa fluorescent dextran specifically labeled the tegument surface (*Figure 1B*), cytoplasmic projections (*Figure 1C*), and the tegumental cell bodies (*Figure 1D*) that sit beneath the parasite's body wall muscles (*Figure 1E,F*). Since isotonic dextran solutions failed to label the tegument, we suspect that specific labeling requires damage to the outer tegumental membranes. Consistent with classic ultrastructural studies, these tegmental cell bodies extend one or more projections towards the tegumental surface (*Morris and Threadgold, 1968*; *Hockley, 1973*) (*Figure 1F*) and appear to form an elaborate interconnected network of cellular projections and cell bodies (*Video 1*). Since the narrowest tegumental cytoplasmic projections are much larger (~100 nm) (*Hockley, 1973*) than the diameter of the fluorescent-dextran conjugate, it is likely that this approach is capable of labeling all cells directly attached to the tegument.

### Definitive tegumental cells express *calpain*, *npp-5*, *annexin* and *gtp-4* but not *tsp-2*

To study the development of the tegument, we next sought to identify molecular markers expressed in tegumental cells and, therefore, performed fluorescence in situ hybridization (FISH) experiments

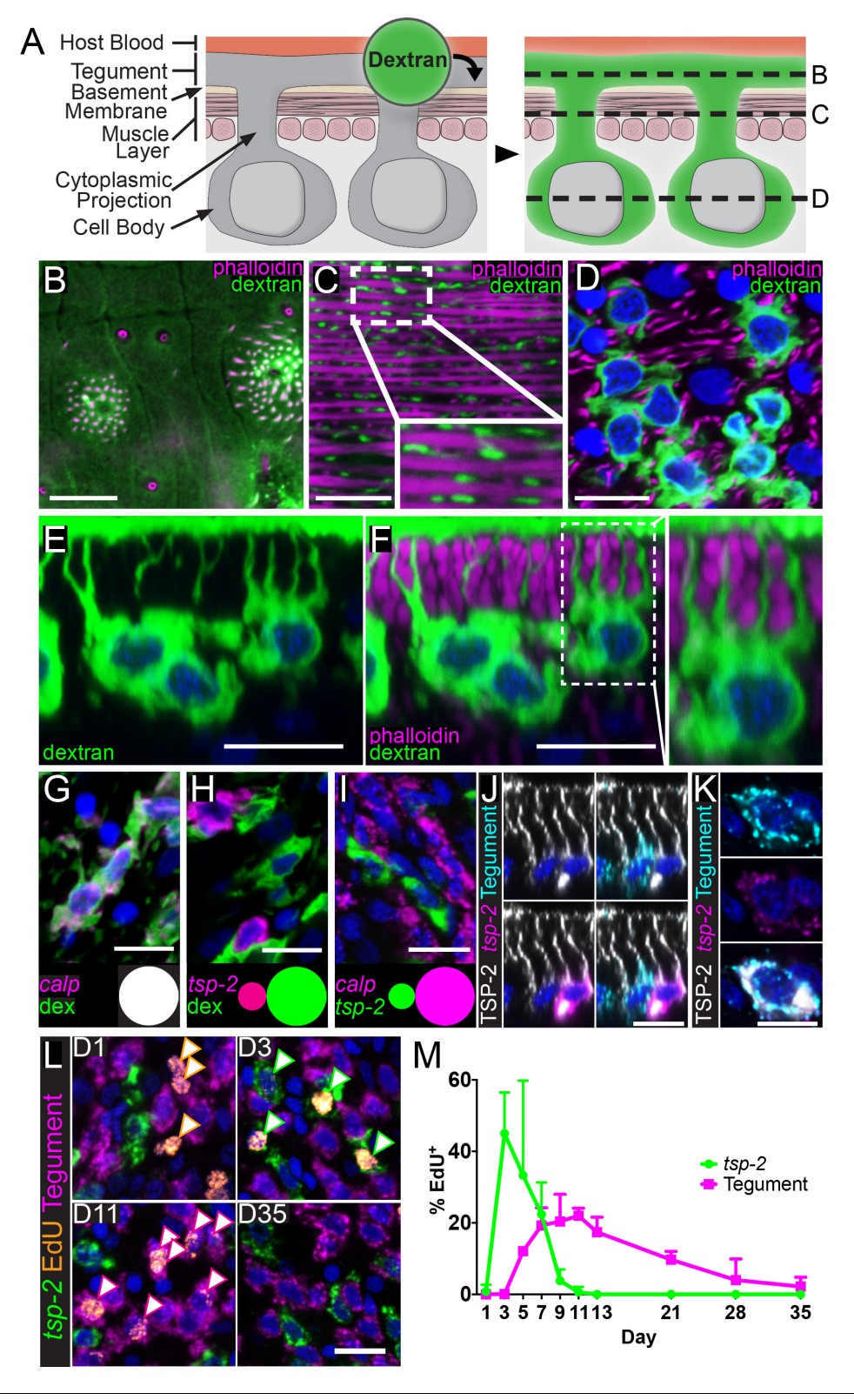

**Figure 1.** *tsp-2*+ neoblast progeny cells fuse with the tegumental syncytium in adult schistosomes. (**A**) Cartoon depicting anatomy of the tegument and fluorescent dextran labeling. (**B–D**) Transverse planes through various levels of the tegument as indicated in (**A**). Phalloidin labels F-actin-rich (**B**) tegumental spines and pores and (**C,D**) muscle fibers; fluorescent dextran labels the tegument, cytoplasmic projections, and tegumental cell bodies. (**E–F**) Side view of dextran-labeled tegument depicting cytoplasmic projections extending from the cell bodies to the surface of the tegument and (**F**)

*Figure 1 continued on next page*

*Figure 1 continued*

intercalating between phalloidin-labeled muscle fibers. (**G–H**) FISH experiments demonstrating the localization of (**G**) *calp* expression (n = 222 cells from three adult male parasites) or (**H**) *tsp-2* expression relative to the dextran-labeled tegumental cells (n = 233 cells from three adult male parasites). Insets show a Venn diagram illustrating the relative overlap of cell populations. (**I**) Double FISH experiment demonstrating the localization of *tsp-2* expression relative to *calp* expression (n = 275 cells from three adult male parasites). (**J**) Immunofluorescence in conjunction with FISH demonstrating that TSP-2 protein is found in both *tsp-2*-expressing cells and in the cells expressing a mixture of tegument markers ('Tegument'). (**K**) Image of a rare *tsp-2* mRNA expressing tegumental cell that is also TSP-2 protein positive. (**L**) EdU pulse-chase experiment examining the kinetics of EdU incorporation into *tsp-2*$^+$ cells and definitive tegumental cells. We find that EdU is incorporated into *tsp-2*$^+$ cells prior to incorporation into cells expressing tegumental markers, consistent with short lived *tsp-2*-expressing progenitors going on to become mature tegumental cells (n = ~130 cells per animal from six adult male parasites per time point). (**M**) Quantification of EdU incorporation in *tsp-2*$^+$ and tegumental cells. Error bars represent 95% confidence intervals. Scale bars: 10 μm.

DOI: https://doi.org/10.7554/eLife.33221.003

The following figure supplements are available for figure 1:

**Figure supplement 1.** FISH examining the expression of several candidate tegument markers.

DOI: https://doi.org/10.7554/eLife.33221.004

**Figure supplement 2.** Examination of TSP-2 protein localization.

DOI: https://doi.org/10.7554/eLife.33221.005

---

on dextran-labeled parasites. Examination of a panel of candidate tegument-specific factors assembled from the literature (*Skelly and Shoemaker, 1996*; *van Balkom et al., 2005*; *Braschi and Wilson, 2006*; *Rofatto et al., 2009*; *Wilson, 2012*) found that mRNAs for *calp*, *npp-5*, *annexin* and *gtp-4* were exclusively expressed in dextran positive cells at the level of the tegument (*Figure 1G* and *Figure 1—figure supplement 1A–C*), suggesting these genes encode markers of tegumental cells. We previously demonstrated that cells expressing the mRNA for the tegument-specific factor *tsp-2* are rapidly produced by neoblasts and then rapidly turned over (*Collins et al., 2016*). Since a variety of proteomic and immunological studies have demonstrated that the TSP-2 protein is associated with the tegument (*van Balkom et al., 2005*; *Braschi and Wilson, 2006*; *Tran et al., 2006*; *Pearson et al., 2012*; *Wilson, 2012*), we were surprised that virtually all *tsp-2* mRNA-expressing cells were dextran-negative despite, in many cases, being found in close proximity to dextran-labeled tegumental cell bodies (*Figure 1H*). Similarly, we did not observe extensive co-localization of *tsp-2* with the tegumental markers *calpain*, *npp-5*, *annexin* and *gtp-4* in adult parasites (*Figure 1I* and *Figure 1—figure supplement 1D–F*). Indeed, extensive examination using both tegumental markers and dextran labeling revealed only five tegumental cells that expressed low levels of *tsp-2* from 3074 tegumental cells examined (~0.2%). We made similar observations with another tegument-enriched factor *sm13* (*Figure 1—figure supplement 1G-L*; 1/1826 tegumental cells was *sm13*$^+$) that is exclusively expressed in *tsp-2*$^+$ cells (*Collins et al., 2016*). Together, these data suggest that *tsp-2* mRNA is not expressed at high-levels in definitive tegumental cells.

## *tsp-2*$^+$ cells include putative progenitors to the definitive tegument

To reconcile the observation that *tsp-2* is not highly expressed in the definitive tegument with the extensive literature linking the TSP-2 protein to the tegument surface, we performed immunofluorescence with an anti-TSP-2 antibody (*Pearson et al., 2012*). We verified the specificity of this antibody by Western-blot following *tsp-2* RNAi treatment (*Figure 1—figure supplement 2A*). Similar to previous studies (*Tran et al., 2006*; *Pearson et al., 2012*), we observed high levels of TSP-2 protein localized on the tegumental surface (*Figure 1—figure supplement 2B,C*). Upon the optimization of labeling conditions, we also noted that TSP-2 protein could be detected in *tsp-2* mRNA-expressing cell bodies and their projections

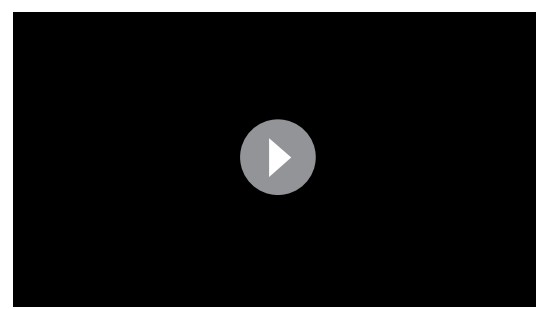

**Video 1.** Movie depicting the elaborate interconnected network of the parasite's tegument and the attached tegumental cell bodies.

DOI: https://doi.org/10.7554/eLife.33221.006

which extend toward the tegument surface (*Figure 1J*, *Figure 1—figure supplement 2D*). We also detected lower levels of TSP-2 in tegumental cell bodies expressing a mixture of tegument-specific mRNAs (*annexin*, *gtp-4*, *npp-5*, and *calp*) (*Figure 1J*) or labeled with dextran (*Figure 1—figure supplement 2E*). Although lower levels of TSP-2 were typically found in tegumental cell bodies, higher levels of the protein were observed on the apical sides of these cells and in the projections extending to the tegument surface (*Figure 1J*, *Figure 1—figure supplement 2D–E*, *Video 2*). Additionally, we observed rare cells expressing markers of definitive tegumental cells, TSP-2 protein, and low levels of *tsp-2* mRNA (*Figure 1K*). Based on these data, an attractive model is that *tsp-2* mRNA-expressing cells include a population of tegumental precursors and that as these cells differentiate to mature tegumental cells, the TSP-2 protein remains stable while the *tsp-2* mRNA is down-regulated.

To explore the model that *tsp-2*$^+$ cells include a population of tegumental precursors, we examined the kinetics of the differentiation of neoblasts to *tsp-2*$^+$ cells and tegumental cells by performing pulse-chase experiments with the thymidine analog 5-Ethynyl-2'-deoxyuridine (EdU). In these experiments, we injected schistosome-infected mice with EdU to label proliferative neoblasts and then examined the kinetics by which these cells differentiate to produce both *tsp-2*$^+$ and definitive tegumental cells. If *tsp-2*$^+$ cells include precursors to the definitive tegument we anticipate: (I) that EdU would chase into the nuclei of *tsp-2*$^+$ cells prior to the definitive tegumental cells and (II) that as EdU signal is lost from the *tsp-2*$^+$ cell compartment we would observe a concomitant increase in the fraction of EdU$^+$ tegumental cells. Consistent with this model, at D3 following an EdU pulse 45% of *tsp-2*$^+$ cells are EdU$^+$, whereas just 0.1% of definitive tegumental cells are EdU$^+$ at this time point. After D3, however, the fraction of EdU$^+$*tsp-2*$^+$ cells began to drop, and the fraction of EdU$^+$ tegumental cells jumped to 12% by D5 before peaking at around 20% between D7 and D11 (*Figure 1L–M*). By D35 the fraction of EdU$^+$ tegumental cells dropped to 2.2%, suggesting that tegumental cells are subject to physiological turnover inside a mammalian host. These data, together with our TSP-2 immunolabeling studies, are consistent with a model in which neoblasts produce a population of short-lived *tsp-2*$^+$ progenitor cells that differentiate and fuse with the tegument. Thus, tegumental cells appear to rely on neoblasts for their continual maintenance.

## FACs purification and molecular characterization of neoblasts and TSP-2$^+$ cells

As a first step towards understanding how tegument development and tissue homeostasis is regulated on a molecular level, we set out to characterize the expression of genes in both neoblasts and *tsp-2*$^+$ cells. Although our previous work exploited the radiation sensitivity of neoblasts and *tsp-2*$^+$ cells to identify candidate cell-type specific markers (*Collins et al., 2013*; *Collins et al., 2016*), we were interested in directly measuring gene expression in these cells. To this end, we developed a FACS methodology to purify both proliferative neoblasts and TSP-2$^+$ tegumental progenitors from single-cell suspensions of schistosome somatic tissues (*Figure 2A*).

Since schistosome neoblasts appear to be the only proliferative somatic cell type (*Collins et al., 2013*), we adapted a methodology developed for FACS purifying neoblasts from free-living planarian flatworms using the live cell DNA-binding dye Hoechst 33342 (*Hayashi et al., 2006*). In this approach, S/G2/M phase neoblasts can be purified from non-cycling (2N DNA content) cells due to their elevated DNA content (>2N) as measured by Hoechst 33342 labeling intensity (*Figure 2A*). Tetraspanins are transmembrane proteins often localized to the cell surface (*Charrin et al., 2014*). Since our anti-TSP-2 antibody is directed to a putative extracellular loop of TSP-2 (*Pearson et al., 2012*), we also employed this antibody to FACS purify TSP-2$^+$ cells (*Figure 2A*). Performing FACS on cell populations labeled with both Hoechst 33342 and anti-TSP-2, we could clearly resolve cells

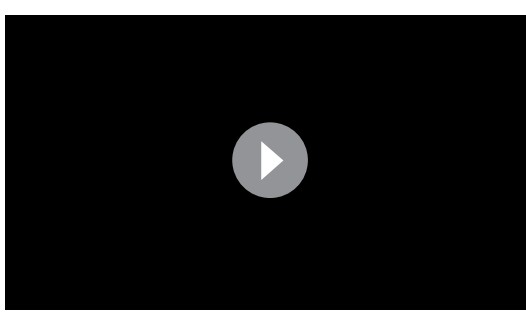

**Video 2.** Movie depicting the localization of TSP-2 protein relative to the tegument and *tsp-2* expressing cells.
DOI: https://doi.org/10.7554/eLife.33221.007

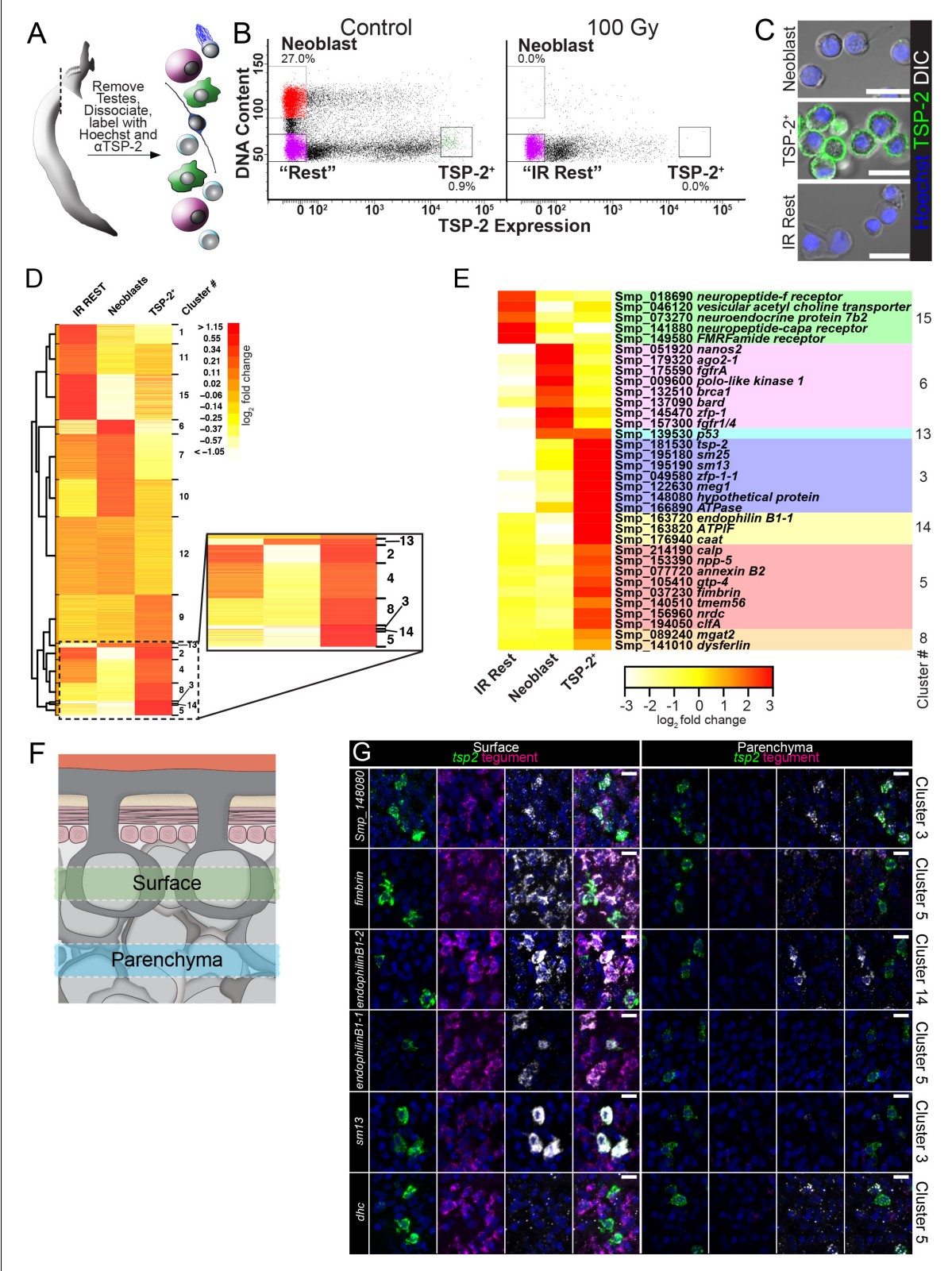

**Figure 2.** FACS purification and transcriptional profiling identifies molecules expressed in neoblasts and cells associated with the tegumental lineage. (A) Cartoon depicting FACS purification strategy. (B) FACS plots showing various cell populations in control and following gamma-irradiation. Percentages represent fraction of the number of cells in the boxed region over the total number of live cells. (C) Confocal micrographs of the sorted cell populations labeled with Hoechst and an anti-TSP-2 antibody. (D) Heatmap showing clustering analysis of genes expressed in the indicated cell

*Figure 2 continued on next page*

*Figure 2 continued*

populations. Inset shows TSP-2 enriched clusters. (E) Heatmap showing the relative expression of individual genes in each cell population. These genes are organized by cluster. (F) Cartoon depicting the approximate regions imaged in panel G. (G) Maximum intensity projection of z-stacks acquired either at superficial levels ('Surface') or deeper in the parasite tissue ('Parenchyma'). The gene expression cluster of each gene examined is listed on the right. Scale bars: 10 μm.

DOI: https://doi.org/10.7554/eLife.33221.008

The following figure supplements are available for figure 2:

**Figure supplement 1.** Microscopic imaging of sorted 'TSP-2 Intermediate' cells.

DOI: https://doi.org/10.7554/eLife.33221.009

**Figure supplement 2.** Examination of the expression of genes expressed in TSP-2-enriched clusters.

DOI: https://doi.org/10.7554/eLife.33221.010

**Figure supplement 3.** Graphical summary of genes expressed in TSP-2-enriched clusters.

DOI: https://doi.org/10.7554/eLife.33221.011

with >2N DNA content (putative neoblasts) and 2N cells with high levels of anti-TSP-2 labeling (*Figure 2B*). Cells with >2N DNA content possessed typical neoblast morphology (small cells with a high nuclear:cytoplasmic ratio), whereas 2N cells with the highest levels of TSP-2$^+$ labeling possessed a lower nuclear:cytoplasmic ratio and labeled strongly for TSP-2 on their surface (*Figure 2C*). We also noted a large population of cells with intermediate levels of TSP-2 labeling (i.e., cells with $10^2$–$10^4$ in relative TSP-2 labeling intensity, *Figure 2B*). Visual examination of these cells found that they did not possess high levels of TSP-2 surface labeling. Instead, these 'TSP-2 Intermediate' cells had either no TSP-2 surface labeling or had pieces of TSP-2-labeled debris attached to their surface (*Figure 2—figure supplement 1*). Since TSP-2 is present at high-levels on the outer tegument, we believe these cells are falsely scored as TSP-2$^+$ due to the contamination of TSP-2$^+$ tegumental debris in our FACS preparations.

To unambiguously confirm the identity of the neoblast and TSP-2$^+$ cell populations, we also performed FACS with parasites 7 days post-treatment with 100 Gy of γ-irradiation, which is sufficient to deplete both neoblasts and *tsp-2*$^+$ cells but spare other differentiated cell types in the worms (*Collins et al., 2013*; *Collins et al., 2016*). Both the neoblasts and TSP-2$^+$ cell populations are eliminated following irradiation, confirming the specificity of our sorting procedure (*Figure 2B*). We also FACS-purified 2N TSP-2$^-$ irradiation insensitive cells, which we refer to hereafter as 'IR Rest' cells (*Figure 2B*). Consistent with the idea that the IR Rest cells represent various differentiated cell types in the parasite, the FACS-purified cells displayed a range of cellular morphologies (e.g., ciliated cells) and nuclear:cytoplasmic ratios (*Figure 2C*).

To define cell-type specific expression profiles, we performed RNAseq on purified neoblasts, TSP-2$^+$ cells, and IR Rest cell populations (*Figure 2B–C*). We performed pair-wise comparisons to define relative differences in gene expression between these three cell populations (*Supplementary file 1*) and used model-based clustering (*Si et al., 2014*) (*Figure 2D*) to identify genes whose expression was specifically enriched in each cell population (*Supplementary file 2*). From this clustering analysis, we found clusters of genes whose expression was enriched to varying degrees in the IR Rest (clusters 1, 11, 15), neoblast (cluster 6 and to a lesser extent 10), and TSP-2$^+$ cell populations (cluster 3, 14, 5, 8). Examination of genes in these clusters identified anticipated cell-type-specific markers: the IR Rest-enriched cluster 15 included genes whose expression is associated with differentiated cells such as neurons (i.e. *neuropeptide f receptor*, *neuroendocrine protein 7b2*); the neoblast-enriched cluster 6 included known neoblast-specific factors including *fgfrA*, *nanos2*, and a variety of cell cycle-associated regulators; and the TSP-2$^+$-enriched clusters 3, 14, 8 included *tsp-2* and a variety of genes previously shown to be expressed in *tsp-2*$^+$ cells including *sm13*, *sm25*, *cationic amino acid transporter*, and *dysferlin* (*Figure 2E*) (*Collins et al., 2016*). We also identified clusters of genes whose expression was enriched in two of the three cell populations. For instance, cluster 13 included genes enriched in both neoblasts and TSP-2$^+$ cells. Among the genes in cluster 13 was the *S. mansoni p53* homolog that was previously demonstrated to be highly expressed in both neoblasts and *tsp-2*$^+$ cells (*Collins et al., 2016*).

Since we found that TSP-2-labled cells expressed tegument-enriched genes (*Figure 1J,K*) we also reasoned that our FACS data might include markers of definitive tegument. Indeed, we noted that the TSP-2-enriched cluster 5 included all four of our validated markers of definitive tegumental cells

(*calp*, *npp-5*, *annexin*, and *gtp-4*) (*Figure 2E*). To explore the significance of this observation, we performed an *in situ* hybridization screen to characterize the expression of genes present in TSP-2-enriched clusters, giving specific attention to genes present in cluster 5. Examining the expression of genes both at the level of the tegument and deeper inside the parenchyma where most *tsp-2*+ cells reside (*Figure 2F*), we found that 26/28 genes in clusters 3, 14, 5, 8 that gave discernable expression patterns were expressed in either *tsp-2*+ cells or definitive tegumental cells (*Figure 2G*, *Figure 2—figure supplements 2* and *3*, *Supplementary file 3*). Among these genes, 15/20 in cluster five alone were expressed in definitive tegumental cells (*Supplementary file 3*), suggesting that genes in this cluster appear to be enriched for tegument-specific transcripts. We also noted from these analyses that *tsp-2*+ cells are heterogeneous on a molecular level: cells deeper in the parenchyma tended to express a dynein heavy chain homolog (*Figure 2G*), whereas more superficial *tsp-2*+ cells expressed *sm13* (*Figure 2G*) and *sm25* (*Figure 2—figure supplement 2*). Similarly, we found a pair of transcripts encoding Endophilin B1 homologs that were expressed at high levels in a subset of mature tegumental cell bodies (*Figure 2G*). This heterogeneity could highlight populations of cells at different stages of commitment to the tegumental lineage. Taken together, these data suggest that clusters 3, 5, 8, 14 are enriched for transcripts expressed in either *tsp-2*+ cells or definitive tegumental cells, providing an additional line of evidence connecting *tsp-2*+ cells and the definitive tegument.

## An RNAi screen identifies *zfp-1* and *zfp-1—1* as potential regulators of tegument development

To define genes that regulate the development of the tegument lineage, we used our neoblast and TSP-2+-enriched datasets to select candidates for an RNAi screen of genes encoding putative transcription factors, RNA binding proteins, signaling molecules, and schistosome-specific proteins. For this screen, we performed RNAi on adult parasites and examined the numbers of neoblasts (by EdU-labeling) and *tsp-2*+ cells (by FISH) (*Figure 3A*). We reasoned that genes required for general neoblast maintenance/proliferation would be essential for the maintenance of both EdU+ neoblasts and *tsp-2*+ cells (e.g., *histone H2B* [*Figure 3B*]), whereas genes important for tegument development would be essential for the maintenance of *tsp-2*+ cells but dispensable for neoblast maintenance (*Figure 3A*). From these experiments, we identified several factors essential for neoblast maintenance, including: a homolog of the human breast cancer type one susceptibility protein (BRCA1), a homolog of the BRCA1 associated RING domain 1 (BARD1) protein, a previously uncharacterized fibroblast growth factor (FGF) receptor, and a homolog of the p53 tumor suppressor (*Figure 3B*). A number of other genes were screened that gave no stem cell or *tsp-2* phenotype (*Figure 3—figure supplement 1*). Given our focus on genes required for tegumental differentiation, these genes were not explored further.

In addition, we found that RNAi of genes encoding two related C2H2 zinc finger proteins, *zfp-1* and *zfp-1-1*, resulted in a reduction in the total number of *tsp-2*+ cells yet spared the number of EdU-labeled neoblasts (*Figure 3C*). Indeed, RNAi-mediated transcript reduction of either *zfp-1* or *zfp-1-1* (*Figure 3—figure supplement 2*) resulted in an approximately 50% reduction in the number of *tsp-2*+ cells (*Figure 3C,D*) and led to no change in the total number of *nanos2*+ neoblasts capable of incorporating EdU (*Figure 3E,F*). The effect of *zfp-1* and *zfp-1-1* RNAi treatment was not specific to the expression of *tsp-2*+, as RNAi of either of these genes similarly led to a sizable decrease in the total number of cells expressing *sm13*, a gene that is expressed in nearly all superficial *tsp-2*+ cells (*Figure 3G,H*, *Figure 3—figure supplement 2*). These observations suggest *zfp-1* and *zfp-1-1* are important for the differentiation and/or maintenance of *tsp-2*+ cells.

Consistent with our RNAseq data, we found that *zfp-1* was expressed exclusively in *nanos2*+ neoblasts and not in *tsp-2*+ cells (*Figure 3I,J*). Conversely, *zfp-1-1* was not expressed in *nanos2*+ neoblasts but was expressed at high levels in *tsp-2*+ cells (*Figure 3K,L*). Similar to other transcripts enriched in *tsp-2*+ cells, *zfp-1-1* appeared to be expressed in a subset of *tsp-2*+ cells that were located more internally within the parasite (*Figure 3M*) but not in more peripherally-located *sm13*+-*tsp-2*+ cells (*Figure 3—figure supplement 3*). Since neoblasts are typically located deeper inside the parasite, these more internal *tsp-2*+*zfp-1-1*+ cells could represent early neoblast progeny, whereas the peripheral *tsp-2*+*sm13*+ cells may represent more mature tegumental progenitors. We further determined that *zfp-1-1* was not expressed in definitive tegumental cells (*Figure 3L*) and that *zfp-1* and *zfp-1-1* were not co-expressed (*Figure 3J*; 0/64 *zfp-1-1*+ cells were *zfp-1*+). Thus, *zfp-

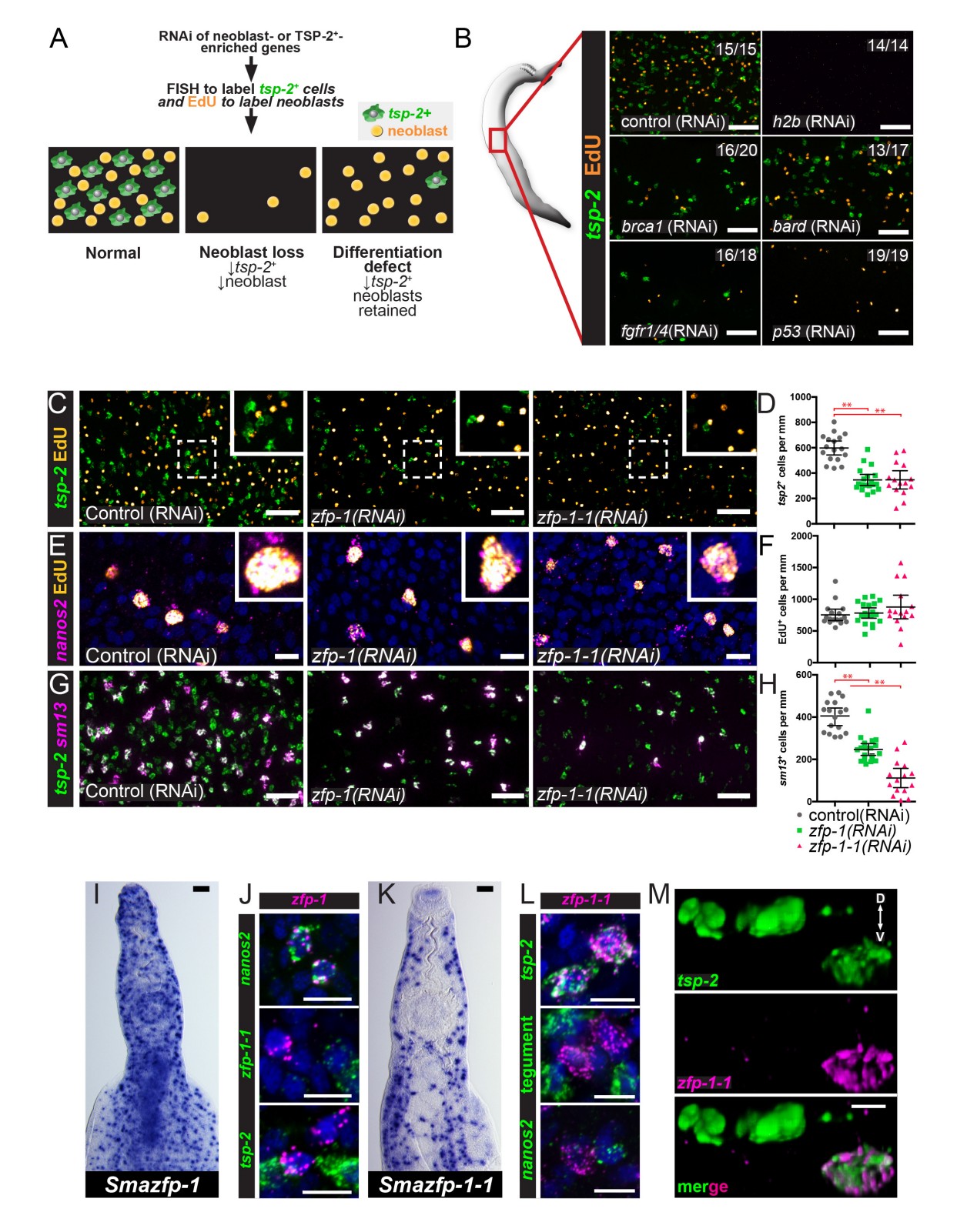

**Figure 3.** An RNAi screen identifies *zfp-1* and *zfp-1–1* as genes required for the production of *tsp-2*⁺ cells. (**A**) Cartoon depicting the RNAi screening strategy used to identify regulators of tegument progenitor specification. Candidate genes were knocked-down using RNAi, worms were pulsed with EdU for 4 hrs and then fixed. Neoblasts and tegument progenitor cells were observed using EdU detection and *tsp-2* RNA FISH, respectively. (**B**) Results of control RNAi experiments. Negative control RNAi preserves *tsp-2* cells and neoblasts whereas *h2b* RNAi results in a loss of neoblasts and

*Figure 3 continued on next page*

*Figure 3 continued*

tsp-2 cells. *brca1*, *bard*, *fgfr1/4*, and *p53* RNAi results in a partial depletion of neoblasts and a proportional decrease in *tsp-2*[+] cells. Representative maximum intensity confocal projections are shown. Numbers represent the fraction of parasites displaying the observed phenotype. (**C**) Maximum intensity projection showing *tsp-2* expression and EdU incorporation in *zfp-1(RNAi)* or *zfp-1-1(RNAi)* parasites. (**D**) Quantification of the number of *tsp-2*[+] cells per mm of worm. Control(RNAi) n = 17, *zfp-1(RNAi)* n = 19, *zfp-1-1(RNAi)* n = 15. (**E**) Maximum intensity projection showing *nanos2* expression and EdU incorporation in *zfp-1(RNAi)* or *zfp-1-1(RNAi)* parasites. (**F**) Quantification of the number of EdU[+] cells per mm worm. Control(RNAi) n = 17, *zfp-1(RNAi)* n = 19, *zfp-1-1(RNAi)* n = 15. (**G**) Maximum intensity projection showing *tsp-2* and *sm13* expression in *zfp-1(RNAi)* or *zfp-1-1(RNAi)* parasites. (**H**) Quantification of the number of *sm13*[+] cells per mm worm. Control(RNAi) n = 17, *zfp-1(RNAi)* n = 19, *zfp-1-1(RNAi)* n = 15. (**I**) WISH showing *zfp-1* expression in adult male worm. (**J**) Double FISH showing expression of *zfp-1* relative to *nanos2* (a neoblast marker), *zfp-1–1*, and *tsp-2*. (**K**) WISH showing *zfp-1–1* expression in adult male worm. (**L**) Double FISH showing expression of *zfp-1–1* relative to *tsp-2*, a mixture tegument-specific markers (tegument), and *nanos2* (a neoblast marker). (**M**) 3D rendering showing expression of *zfp-1–1* in a subset of *tsp-2*[+] cells. The dorsal and ventral surfaces of the animal are oriented towards the top and the bottom of the image, respectively, as indicated by the arrows in the first panel. Scale bars: B, C, G, I, K 50 µm; E, J, L, M 10 µm. Error bars represent 95% confidence intervals, **p<0.01 (Student's t-test).

DOI: https://doi.org/10.7554/eLife.33221.012

The following figure supplements are available for figure 3:

**Figure supplement 1.** RNAi screen of candidate tegument biogenesis regulators.

DOI: https://doi.org/10.7554/eLife.33221.013

**Figure supplement 2.** Quantification of gene expression in *zfp-1(RNAi)* and *zfp-1-1(RNAi)* parasites.

DOI: https://doi.org/10.7554/eLife.33221.014

**Figure supplement 3.** Gene expression patterns of *sm13* and *zfp-1–1*.

DOI: https://doi.org/10.7554/eLife.33221.015

*1* expression appears to be neoblast-specific, whereas *zfp-1–1* expression is enriched in a subset of *tsp-2*[+] cells.

## *zfp-1* and *zfp-1–1* are members of a family of flatworm-specific DNA binding proteins whose homolog in planarians regulates epidermal lineage specification

We examined the amino acid sequences of the proteins encoded by *zfp-1* and *zfp-1–1*. Not only were the three C2H2 zinc finger domains of ZFP-1 and ZFP-1–1 highly similar to one another, but we also uncovered closely-related C2H2 zinc finger domain-containing proteins in the genomes of free-living (i.e., planarians and macrostomids) and parasitic flatworms (i.e., flukes, tapeworms, monogeneans) (*Figure 4A*). A thorough examination of proteins from taxa outside the Platyhelminthes failed to find any close relatives that shared both high sequence identity and a similar number of C2H2 domains, suggesting that these proteins are likely to be flatworm-specific. Phylogenetic analysis of these proteins revealed two distinct groups of these ZFP-1 family proteins: one group more similar to the schistosome *zfp-1* and another more closely related to *zfp-1–1* (*Figure 4B*). Among the homologs identified was a protein encoded from the *zfp-1* gene in the planarian *Schmidtea mediterranea*. In parallel to our model for tegument renewal by short-lived *tsp-2*[+] tegumental progenitors, the planarian epidermis is perpetually rejuvenated from a population of short-lived epidermal progenitors derived from the neoblasts (*Eisenhoffer et al., 2008*; *van Wolfswinkel et al., 2014*). The production of these epidermal progenitors relies on the planarian *zfp-1*, which is expressed in a subset of lineage-restricted neoblasts (*van Wolfswinkel et al., 2014*). Thus, our results with *zfp-1* and *zfp-1–1* suggest the potential for a conserved role for these proteins in coordinating epidermal biogenesis programs among flatworms.

Although *zfp-1* has been previously characterized in *S. mediterranea,* the molecular function of this group of novel proteins is not clear. Since we found proteins in this family shared little homology outside the three C2H2 zinc finger domains, we reasoned that these domains are likely key to the function of these proteins. C2H2 zinc finger domains are best known for their ability to function as transcriptional regulators by binding DNA, however, these domains can also participate in RNA-binding and protein:protein interactions (*Krishna et al., 2003*; *Hall, 2005*; *Brayer and Segal, 2008*). Thus, we examined the sequences of these proteins in more detail. C2H2 zinc finger domains contain two conserved cysteines and two conserved histidines for zinc-binding (highlighted in black background in *Figure 4A*). For the ZFP-1 family proteins, we observed that the residues between the second zinc-coordinating cysteine and the first zinc-coordinating histidine of the second and

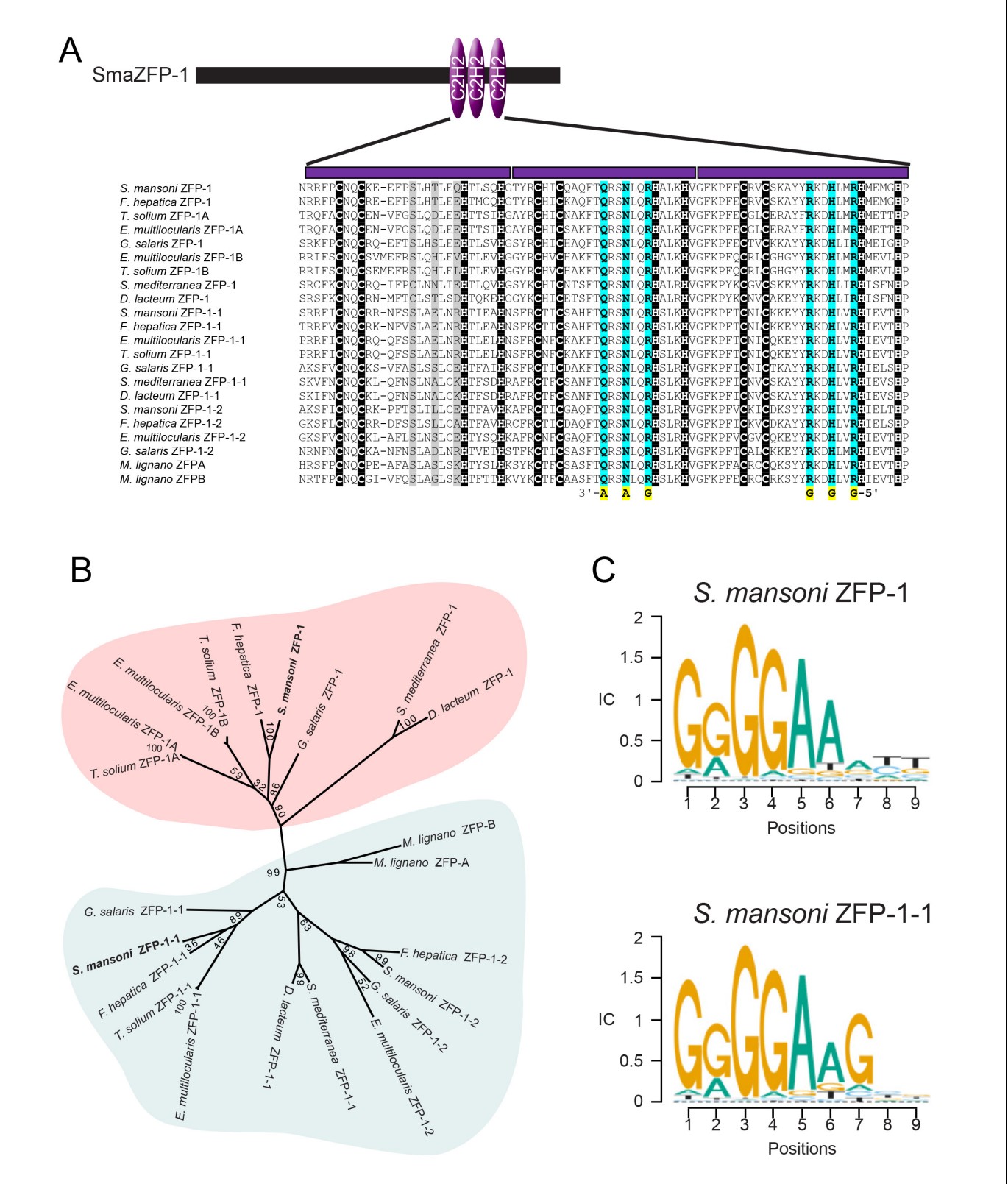

**Figure 4.** ZFP-1 and ZFP-1–1 are flatworm specific zinc finger proteins and are putative transcriptional regulators. (**A**) Multiple protein sequence alignment of the C2H2 domain of several *zfp-1* and *zfp-1–1* homologs. Zinc coordinating residues are shown in black background. Conserved residues contributing to high specificity DNA-binding are highlighted in cyan for the second and third zinc fingers, with the specific DNA base shown below the residue highlighted in yellow. The corresponding positions in the first zinc finger are shown in grey background. The positions determining DNA

*Figure 4 continued on next page*

*Figure 4 continued*
binding specificity in the first zinc finger (highlighted in grey background) either are not well conserved among these proteins or do not contribute to high specificity of DNA binding. (B) Un-rooted phylogenic tree of *ZFP-1* and *ZFP-1–1* homologs from multiple species of flatworms. Numbers at the nodes represent bootstrap values. (C) Predicted DNA binding motif of *zfp-1* and *zfp-1–1* of *S. mansoni* by the ZFModels server.
DOI: https://doi.org/10.7554/eLife.33221.016

third zinc fingers exhibited high sequence conservation, forming the motifs QRSNLQR and RKDHLxR, respectively (*Figure 4A*). Typically, each C2H2 zinc finger interacts with three consecutive DNA base pairs, and the first, fourth, and seventh positions in these motifs (highlighted in cyan background in *Figure 4A*) are key contributors to the binding specificity of the 3'base, the middle base, and 5' base of the primary interaction DNA strand, respectively (*Wolfe et al., 2000*; *Klug, 2010*). Given this stereotypical binding, it is possible to predict target DNA binding sequences solely from amino acid sequences (*Gupta et al., 2014*). Using this model, we predict the that the common preferred DNA binding sequence for all ZFP-1 homologs examined is 5'-GGGGAA-3' (*Figure 4C*), based on the sequence conservation of the last two zinc fingers. Given the highly conserved nature of the residues that contribute to sequence-specific binding, we believe that ZFP-1 family proteins function by binding DNA and presumably act as transcription factors.

## *zfp-1–1* appears to be specifically required for the production of new tegumental cells

If *tsp-2*$^+$ cells are tegumental precursors, and *zfp-1* and *zfp-1–1* play a role in the specification of tegumental cells, we would anticipate that loss of *tsp-2*$^+$ cells following reduction in *zfp-1* and *zfp-1–1* levels would block the birth of new tegumental cells. Eventually the reduction in tegumental cell birth would result in the reduction in the total number of tegumental cells. To determine if this was the case, we knocked down *zfp-1* or *zfp-1–1* and performed an EdU pulse-chase experiment examining the ability to produce new tegumental cells (*Figure 5A*). Following *zfp-1* RNAi treatment, we noted a relatively small, but statistically significant, reduction in the percentage of tegumental cells that were EdU$^+$ (*Figure 5B,C*). In contrast to *zfp-1* RNAi treatment, knockdown of *zfp-1–1* led to an almost complete block in the ability of new cells to be added to the tegument (*Figure 5B,C*). Consistent with these observed reductions in production of new mature tegumental cells, we also noted that RNAi of *zfp-1* or *zfp-1–1* led to 15 and 30 percent reductions in the total density of tegumental cell bodies, respectively (*Figure 5B,C*). Together these data indicate that both zinc finger proteins are important for tegument specification, but that *zfp-1–1* appears to play a more substantial role in the process.

We next sought to determine if loss of *zfp-1* or *zfp-1–1* led to general defects in the ability of parasites to generate non-tegumental lineages. We first monitored the production of new gut cells using the gut-specific marker *cathepsin B*. Like the tegument, the gut is a syncytium, and gut cells appear to be renewed at a relatively high rate (*Collins et al., 2013*; *Collins et al., 2016*). Following a 7 day EdU chase period, we noted that *zfp-1-1(RNAi)* parasites generated new gut cells at roughly the same rate as control-treated worms (*Figure 5D*). Conversely, the rate of new gut cell birth was severely reduced in *zfp-1(RNAi)* worms, suggesting a role for *zfp-1* not just in tegumental differentiation but also in the generation of new gut cells. Given the paucity of cell-type specific markers in schistosomes we next wanted to monitor the general differentiation potential of neoblasts in *zfp-1 (RNAi) and zfp-1-1(RNAi)* parasites. After a 4 hr EdU pulse >95% of EdU$^+$ cells are *nanos2*$^+$ (160/166 EdU$^+$ nuclei, n = 9 male parasites, *Figure 3E*); therefore, we reasoned that we could monitor the general differentiation potential of neoblasts by examining the amount of EdU-labeled nuclei exiting the *nanos2*$^+$ neoblast compartment after a 7 day chase period (*Figure 5A*). However, since *tsp-2*$^+$ cells are the major output of neoblasts (*Collins et al., 2016*), and neither *zfp-1* nor *zfp-1–1* RNAi treatments completely depleted the *tsp-2*$^+$ cell pool, we specifically examined the appearance of EdU$^+$*nanos2*$^-$*tsp-2*$^-$ cells in the parenchyma after a 7 day chase in order to exclude cells related to the tegument lineage. While we noted large numbers of EdU$^+$*nanos2*$^-$*tsp-2*$^-$ cells in both *zfp-1* and *zfp-1–1* RNAi treated parasites, *zfp-1(RNAi)* worms displayed a slight reduction in the total number of EdU$^+$*nanos2*$^-$*tsp-2*$^-$ cells relative to controls (*Figure 5E*). These data suggest that *zfp-1* may play a

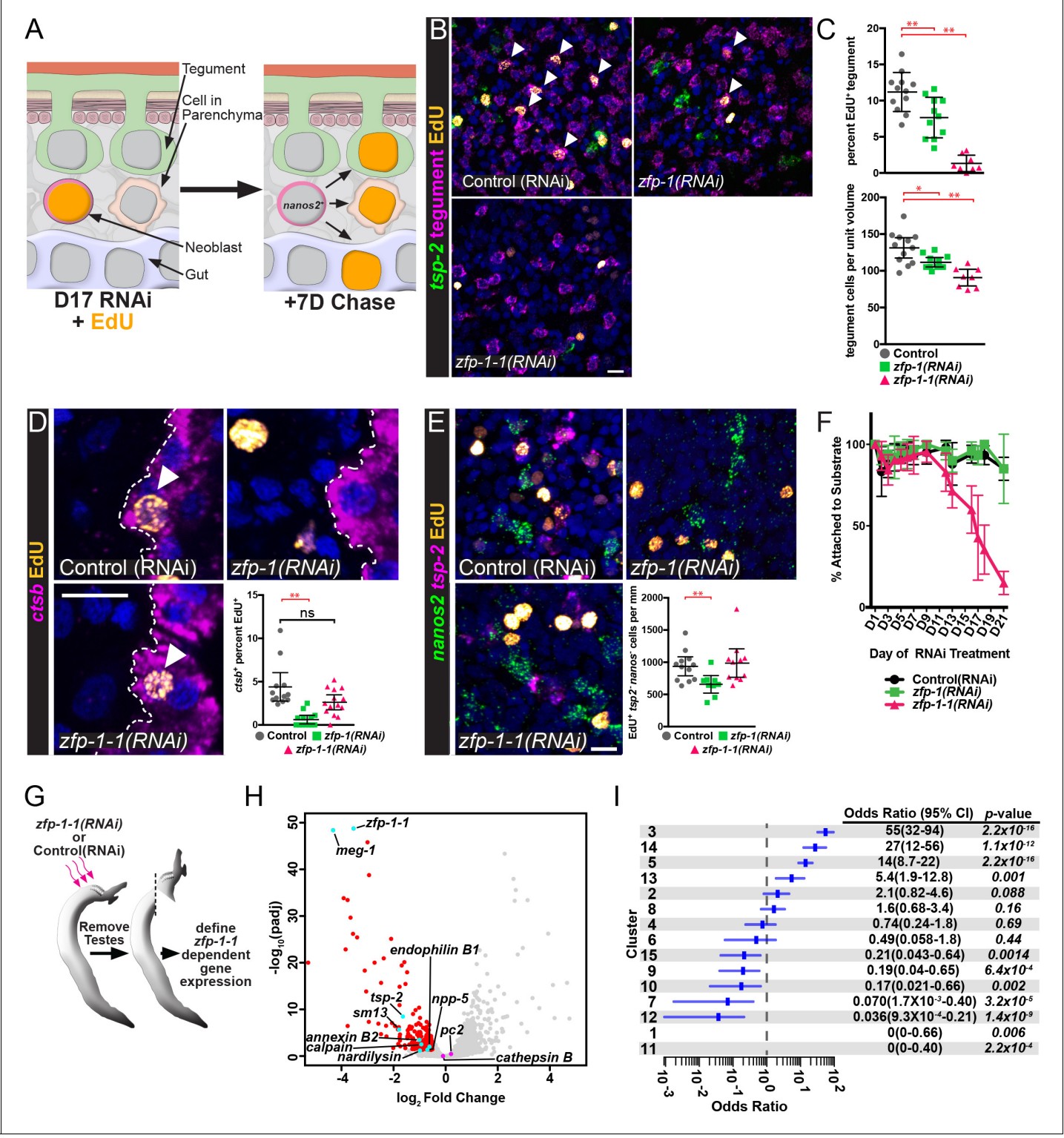

**Figure 5.** ZFP-1 family proteins are required for the production of new tegumental cells. (A) Cartoon depicting the strategy for fate-mapping by EdU pulse-chase experiments. (B) FISH for *tsp-2* and tegumental markers with EdU detection in *zfp-1(RNAi)* or *zfp-1-1(RNAi)* parasites at day seven following an EdU pulse. Arrows represent EdU+ tegumental cells. (C) (Top) Quantification of the percentage of tegumental cells that are EdU+ following a 7 day chase period and (Bottom) tegumental cell density in *zfp-1(RNAi)* or *zfp-1-1(RNAi)* parasites. Control(RNAi) n = 12, *zfp-1(RNAi)* n = 11, *zfp-1-1(RNAi)* n = 8. (D) FISH for *cathepsin B* and EdU detection in *zfp-1(RNAi)* or *zfp-1-1(RNAi)* parasites at day seven following an EdU pulse. Plot represents the percentage of *cathepsin B* + cells that are EdU+. Control(RNAi) n = 12, *zfp-1(RNAi)* n = 13, *zfp-1-1(RNAi)* n = 14. (E) FISH for *nanos2* and *tsp-2* with EdU

*Figure 5 continued on next page*

*Figure 5 continued*

detection in *zfp-1(RNAi)* or *zfp-1-1(RNAi)* parasites at day seven following an EdU pulse. Plot represents the number of *tsp-2⁻* EdU⁺ differentiated cells (i.e., *nanos2⁻* cells) per mm of parasite length. Control(RNAi) n = 12, *zfp-1(RNAi)* n = 10, *zfp-1-1(RNAi)* n = 11. (F) Percentage of the parasites that remain attached to the culture dish at the indicated time point following the first RNAi treatment. n = 5 experiments with approximately 10 worms per RNAi treatment in each experiment. (G) Cartoon depicting strategy for examining transcriptional changes following *zfp–1–1* RNAi. (H) Volcano plot showing differentially expressed genes in *zfp-1-1(RNAi)* worms. Red dots represent genes that are down regulated (−0.5 $\log_2$ fold change, $p_{adj}$ <0.05) in *zfp-1-1* (*RNAi*) worms. Cyan dots indicate genes known to be expressed in the tegument lineage. Magenta dots indicate genes validated to be expressed in differentiated cells. (I) Plot showing odds-ratio (i.e., the relative over- or under-representation) of genes from gene expression clusters among genes down regulated following *zfp-1–1* RNAi. Blue rectangles depict the odds-ratio from a Fisher's Exact Test, whereas blue lines indicate the 95% confidence intervals. Values of odds-ratio and p-values for Fisher's Exact Test shown to right. No genes from expression clusters 1 or 11 were down-regulated following *zfp-1–1* RNAi, so no odds ratio was calculated. From these data, genes from expression clusters 3, 5, 13 and 14 are over-represented (p<0.05), whereas genes from clusters 1, 7, 9, 11, 12, and 15 appear under-represented. Scale bars: 10 μm. Error bars in (C–E) represent 95% confidence intervals, error bars in (F) represent standard deviation. *p<0.05; **p<0.01; ns, not significant (Student's t-test).

DOI: https://doi.org/10.7554/eLife.33221.017

more general role in neoblast differentiation, whereas *zfp-1–1* appears to play a more specific role in the production of new tegumental cells.

During in vitro culture schistosomes use their ventral sucker to attach themselves to the bottom of their cell culture dish (*Collins and Collins, 2016*). In parallel to our observations with *zfp-1–1* in tegumental differentiation, we noted that *zfp-1-1(RNAi)* parasites detached from their culture vessel during RNAi treatment (*Figure 5F*); a similar phenotype was not observed for either control(RNAi) or *zfp-1(RNAi)* animals (*Figure 5F*). These data suggest that loss of tegument cell body density following *zfp-1–1* RNAi may result in gross physical deficits during in vitro culture.

To explore the effects of *zfp-1–1* RNAi in more detail, we performed transcriptional profiling of *zfp-1-1(RNAi)* parasites using RNAseq (*Figure 5G*). As anticipated, RNAi of *zfp-1–1* resulted in reduced expression of transcripts expressed in *tsp-2⁺* cells including *tsp-2*, *meg-1*, and *sm13* (*Figure 5G*, *Supplementary file 4*). Consistent with the observed reduction in the total number of tegumental cells following *zfp-1-1(RNAi)* (*Figure 5C*), we also found that transcripts for the definitive tegumental markers *calpain*, *annexin*, and *npp-5* were significantly down-regulated in *zfp-1-1(RNAi)* parasites (*Figure 5H*, *Supplementary file 4*). Importantly, we did not observe significant changes in the expression of genes associated with the schistosome nervous system (e.g., *pc2* (*Protasio et al., 2017*)) nor in genes associated with the intestine (*cathepsin B*) in *zfp-1-1(RNAi)* parasites (*Figure 5H*). To further explore the specificity of *zfp-1–1* RNAi for cells within the tegument lineage, we examined if genes represented by each of our individual expression clusters (*Figure 2D*) were statistically-enriched among genes down-regulated in *zfp-1-1(RNAi)* parasites. If the effects of *zfp-1–1* depletion are largely restricted to the tegumental lineage and not to other tissues, we would anticipate that a majority of genes down-regulated in *zfp-1-1(RNAi)* parasites would represent genes expressed in the tegumental lineage. Consistent with this model, we found that clusters of genes with high-levels of TSP-2-enrichment (i.e., 3, 14, 5, and 13) were statistically overrepresented among genes down-regulated ($\log_2$ fold change <−0.5, padj <0.05) following *zfp-1-1(RNAi)* (*Figure 5I*). Conversely, clusters with low-levels of *tsp-2* enrichment (i.e., 1, 11, 7, 12, and 15) were statistically underrepresented among genes down-regulated following *zfp-1-1(RNAi)* (*Figure 5I*). Given these data, and our pulse-chase experiments (*Figure 5B–E*), the effects of *zfp-1–1* RNAi appear to predominantly affect the maintenance of tegumental cells and their progenitors. Therefore, we suggest that *zfp-1–1* represents a critical and specific regulator of tegumental specification in schistosomes.

## Discussion

Here, we describe a novel methodology to fluorescently label the schistosome tegument and its associated cell bodies. Using this labeling approach, we defined cell-type specific markers of the tegument, and together with EdU pulse-chase experiments and immunolabeling for TSP-2, we suggest that *tsp-2⁺* cells contribute to the schistosome tegument. Based on our observations we propose a model in which neoblasts specify cells expressing *tsp-2* that migrate through the mesenchyme. As these progenitors approach the tegument, they extend cellular projections that traverse the muscle layers and basement membranes, and ultimately fuse with the outer tegument

(*Figure 6*). Since we find that tegumental cell bodies are subject to physiological cell turnover (*Figure 1L,M*), and that ablation of tegmental progenitors by *zfp-1* of *zfp-1–1* RNAi results in reduced tegumental cell density (*Figure 5B,C*), it appears that neoblast-driven tegument renewal is essential for the homoeostatic maintenance of tegumental cell number.

One outstanding question relates to the molecular composition of cells within the tegumental lineage. Our data suggest that *tsp-2*⁺ cells contribute to the tegument, but it is not clear if this property extends to all *tsp-2*-expressing cells. Analysis of genes expressed in FACS-purified TSP-2⁺ cells found that several genes were expressed in subsets of *tsp-2*⁺ cells (*Figure 2G*, *Figure 2— figure supplement 2*). One possible interpretation of these observations is that these distinct *tsp-2*⁺ populations represent cells at different stages of commitment to a tegumental fate. However, it is possible that certain subsets of *tsp-2*⁺ are destined to generate other non-tegumental lineages. Interestingly, we also observed that a pair of Endophillin B1-encoding genes are expressed in a subset of mature tegumental cells (*Figure 2G*), opening up the possibility that the tegument is comprised of molecular and functionally distinct cell bodies, despite being a syncytium. Based on the relative distribution of tegument-specific cytoplasmic inclusions, early ultrastructural studies hinted at the possibility that multiple classes of tegumental cell types exist (*Morris and Threadgold, 1968*). Given this possibility, different types of *tsp-2*⁺ cells may give rise to different classes of tegumental cell bodies. Alternatively, a mechanism for tegument cell renewal independent of *tsp-2*⁺ cells may also exist. Detailed studies of these various cell populations using emerging single cell RNA sequencing technology are expected to improve our understanding of this cellular heterogeneity and how it relates to tegument biogenesis.

Although both *zfp-1* and *zfp-1–1* are essential for the normal production of tegumental cells, depletion of *zfp-1–1* appears to have a more profound effect on this process (*Figure 5B,C*). This observation is curious since *tsp-2*⁺ cells are depleted to a similar extent in either *zfp-1(RNAi)* or *zfp-1-1(RNAi)* parasites (*Figure 3C,D*). However, we did note that *zfp-1-1(RNAi)* resulted in a much greater depletion of cells expressing *sm13* compared to *zfp-1 (RNAi)* (*Figure 3G,H*). One possible explanation of this observation is that *zfp-1* and *zfp-1– 1* RNAi treatments have differential effects on cells within the *tsp-2*⁺ compartment. Perhaps *zfp-1* acts in the stem cells to specify early tegumental *tsp-2*⁺ progenitors, whereas *zfp-1–1* acts in early progenitors to control the fate of cells later during the commitment process. Given the effects of *zfp-1–1* on *sm13*⁺ cells,

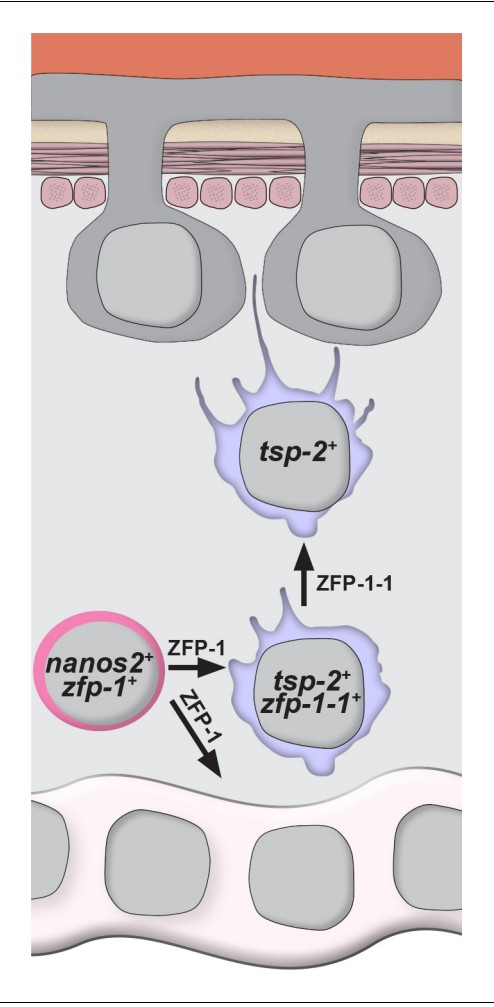

**Figure 6.** Model for the specification of new tegumental cells from neoblasts. Neoblasts (magenta cells) expressing *nanos2* and *zfp-1* specify large numbers of *tsp-2*⁺ cells. Some fraction of *tsp-2* cells express *zfp-1–1*. Within this *tsp-2* compartment are cells that extend cytoplasmic projections ultimately fusing with the tegumental syncytium. Loss of *zfp-1* function results in a general differentiation defect (i.e. loss of both tegument progenitors and gut cells) whereas loss of *zfp-1–1* function results in a specific loss of *tsp-2*⁺ cells responsible for replenishing the tegument. In both cases, depletion of *tsp-2*⁺ cells causes a reduction in the total number of tegumental cell bodies.

DOI: https://doi.org/10.7554/eLife.33221.018

and the location of these cells towards the parasite's surface (*Figure 2G*), it is possible that *sm13*+ cells may represent a population of late tegumental progenitors. A more detailed examination of the various cell types within the *tsp-2*+ compartment is expected to bring clarity to this issue.

In addition to the differential effect on *sm13*+ cells, we found that *zfp-1–1* RNAi treatment resulted in a gradual detachment of the parasite from their culture vessel (*Figure 5F*). Parasites rely upon their ventral sucker to attach to blood vessel walls in the host and to the bottom of culture vessels during *in vitro* culture. As the only part of the worm that physically attaches to solid substrate, one might expect the ventral sucker to experience more 'wear and tear' than the rest of the organism. Like the rest of the worm, the sucker is covered in tegument. While we cannot say that the detachment phenotype is a direct result of the disruption of tegument maintenance, an attractive hypothesis is that the gross effects of loss of tegument cell renewal are first experienced by the sucker in the form of the inability to attach to substrate. Indeed, this hypothesis is supported by the observation that the effects of *zfp-1-1(RNAi)* are largely limited to tegumental cell populations (*Figure 5H–I*). Future studies exploring the function of *zfp-1–1* in the context of host infection could provide important insights into the role for tegmental renewal in parasite survival *in vivo*.

Our data highlight fundamental similarities in the cellular organization of epidermal lineages between schistosomes and the free-living planarian flatworms. Unlike schistosomes, free-living flatworms (e.g., planarians) possess a simple epidermis comprised of a single layer of epithelial cells that rests upon a basement membrane and several layers of muscles (*Hyman, 1951*; *Tyler and Tyler, 1997*; *Tyler and Hooge, 2004*). Counter to the epidermal maintenance strategies of other long-lived metazoa (e.g., cnidarians (*Buzgariu et al., 2015*) and mammals (*Watt, 2001*)), where resident stem cells support the renewal of worn out or damaged epithelial cells, the planarian epidermis is unique as it is completely devoid of proliferative cells (*Newmark and Sánchez Alvarado, 2000*). To fulfill a constant demand for new epidermal cells, neoblasts in planarians specify large numbers of post-mitotic epidermal progenitor cells (*Newmark and Sánchez Alvarado, 2000*; *Eisenhoffer et al., 2008*; *van Wolfswinkel et al., 2014*). In many ways, these epidermal progenitors are similar to *tsp-2*+ tegumental progenitors: they appear to be the primary output of neoblasts, they are rapidly lost following neoblast ablation, and they express a variety of species-specific factors. Furthermore, like schistosomes, these progenitors migrate through the mesenchyme, traverse the muscles and basement membrane, and incorporate into the existing epithelium (*Newmark and Sánchez Alvarado, 2000*). Thus, the cellular organization of epidermal maintenance lineages in planarians and schistosomes appears to be quite similar despite resulting in two very different tissues (epidermis vs. tegument).

In addition to the similarities in their cellular organization, our data, together with previous studies of planarians (*Wagner et al., 2012*; *van Wolfswinkel et al., 2014*), suggest that flatworm epidermal lineages also rely on members of the *zfp-1* family of flatworm-specific transcriptional regulators. Despite the apparent conserved function of these regulators, we do note some differences in the function of *zfp-1* family proteins in planarians and schistosomes. The planarian and schistosome *zfp-1* genes are both expressed in neoblasts, and based on sequence similarity they appear to be orthologous (*Figure 4B*). However, the planarian protein is specifically required for the maintenance of the epidermal lineage, whereas the schistosome protein appears to be essential for both tegumental and non-tegumental lineages (*Figure 5D*). Thus, it would appear the schistosome *zfp-1* homolog plays a more general role in cellular differentiation. These observations, however, do not rule out possibility that the schistosome *zfp-1* protein is directly responsible for specifying tegument fates. Indeed, loss of the non-tegumental lineages following *zfp-1* RNAi could represent a compensatory mechanism by the neoblasts to fulfill a high demand for new tegumental cells. Although a specific role for *zfp-1* cannot be demonstrated at this time, the schistosome *zfp-1–1* appears to have a specific role in tegumental fates. Like *zfp-1*, the schistosome *zfp-1–1* has a related homolog in planarians (*Figure 4B*). While this planarian homolog has not been characterized, recent single cell transcriptional analyses suggest that the expression of this gene is enriched in the epidermal lineage (*Wurtzel et al., 2015*). Clearly, more detailed studies of these zinc finger proteins, and their roles in epidermal development, in both free-living and parasitic flatworms are essential to determine the significance of these observations.

Given these apparent similarities between planarians and schistosomes, and a wealth of evidence indicating that the Neodermata are descendants of free-living flatworms (*Ehlers, 1985*; *Egger et al., 2015*; *Laumer et al., 2015*), it is possible that the evolution of the tegument, and perhaps even the

emergence of parasitism, has its roots in the epidermal biogenesis programs of the free-living ancestors to modern day Neodermata. By modulating the basic cellular behaviors of epidermal progenitor cells during the course of evolution, perhaps there was a shift from migratory epidermal progenitors that intercalate into the multi-cellular epithelium to fusogenic progenitor cells that give rise to the syncytial tegument. Given this model, we suspect that our observations of neoblast-driven tegument biogenesis in schistosomes are likely to extend to all members of the Neodermata. Therefore, further study of tegumental development is expected to provide clues relevant for understanding the evolutionary forces that gave rise to parasitism in flatworms. Furthermore, since the tegument is critical to parasite biology, understanding the tegument lineage, and the molecular targets of *zfp-1* homologues, in diverse flatworms could suggest novel therapies to blunt tegument development in this important group of parasites.

## Materials and methods

### Parasite acquisition and culture

Adult *S. mansoni* (6–7 weeks post-infection) were obtained from infected female mice by hepatic portal vein perfusion with 37°C DMEM (Sigma-Aldrich, St. Louis, MO) plus 10% Serum (either Fetal Calf Serum or Horse Serum) and heparin. Parasites were cultured as previously described (*Collins et al., 2016*). Unless otherwise noted, all experiments were performed with male parasites.

### RNA interference

For *tsp-2* RNAi experiments, 10 freshly perfused male parasites (either as single worms or paired with females) were treated with 20 µg/ml dsRNA for 3 days in Basch Media 169. dsRNA was generated by in vitro transcription and was replaced every day. On the 3rd day, the worms were given fresh media. Thereafter, every 3 days the worms received fresh media and 20 µg/ml dsRNA for a total of 28 days and then the parasites were fixed as previously described (*Collins et al., 2013*). For the candidate RNAi screen, 10 freshly perfused male parasites (either as single worms or paired with females) were treated with 30 µg/ml dsRNA for 7 days in Basch Media 169. dsRNA was generated by in vitro transcription and was replaced every day. On the eighth day, the worms were given fresh media. Thereafter, every fourth day the worms received 60 µg/ml dsRNA (~24 hr of exposure to dsRNA before the media was changed) for a total of 17 days. On day 17, the worms were pulsed with 10 µM EdU for 4 hr before being fixed as previously described (*Collins et al., 2013*). The candidate screen was performed twice.

For EdU pulse-chase RNAi experiments, 10 freshly perfused male parasites (either as single worms or paired with females) were treated with 30 µg/ml dsRNA for 7 days in Basch Media 169. dsRNA was generated by in vitro transcription and was replaced every day. On the eighth day, the worms were given fresh media. Thereafter, every fourth day the worms received 60 µg/ml dsRNA (~24 hr of exposure to dsRNA before the media was changed) for a total of 28 days. On day 21, the worms were pulsed with 10 µM EdU for 4 hr after which the media was changed. On day 28, the worms were fixed as previously described (*Collins et al., 2013*).

As a negative control for RNAi experiments, we used a non-specific dsRNA containing two bacterial genes (*Collins et al., 2010*). cDNAs used for RNAi and in situ hybridization analyses were cloned as previously described (*Collins et al., 2010*); oligonucleotide primer sequences are listed in *Supplementary file 5*. Quantitative PCR analyses to examine knockdown efficiency were performed as previously described (*Collins et al., 2013*; *Collins et al., 2016*).

### Parasite labeling and imaging

Colorimetric and fluorescence in situ hybridization analyses were performed as previously described (*Collins et al., 2013*; *Collins et al., 2016*). To strongly label the entire cytoplasm of tegumental cells by FISH, in some instances we pooled probes recognizing the tegument-specific markers *calpain, gtp-4, annexin, and npp-5*. For dextran labeling, freshly perfused worms were collected in the bottom of a 15 ml conical tube, all residual media was removed, and 100 µl of 5 mg/ml solution of biotin-TAMRA-dextran (Life Technologies D3312) dissolved in ultrapure water was added to ~50 parasites. These worms were constantly agitated by gentle vortexing for 3–4 min, and then doused with 10 ml of fixative solution (4% formaldehyde in PBSTx (PBS + 0.3% triton-X100)) to stop the

labeling. The fixative solution was removed and replaced with 10 ml of fresh fixative solution to dilute residual dextran. The worms were fixed for 4 hr in the dark with mild agitation. Worms were then washed with 10 ml of fresh PBSTx for 10 min. Dextran-labeled worms were then labeled with Alexa Fluor 488-conjugated phalloidin (Lifetech A12379) (1:40 dilution in 1% bovine serum albumin in PBSTx) overnight or dehydrated in methanol and processed for in situ hybridization or immunofluorescence. In vivo and in vitro EdU labeling and detection experiments were performed as previously described (*Collins et al., 2013*). However, for the 5 week in vivo EdU pulse-chase experiments, mice were only exposed to ~30 cercariae to assure the mice would not succumb to schistosome infection prior to the end of the experiment. For immunofluorescence, worms processed for in situ hybridization or dextran labeling were incubated in blocking solution (0.1 M Tris pH 7.5, 0.15 M NaCl and 0.1% Tween-20 with 5% Horse Serum and 0.5% Roche Western Blocking Reagent (*King and Newmark, 2013*)) for 1 hr at room temperature and incubated overnight in affinity purified anti-TSP-2 (*Pearson et al., 2012*) diluted 1:1000 in blocking solution at 4°C. The following day samples were washed 6 × 20 m in PBSTx, incubated overnight in a 1:1000 dilution of AlexaFluor 488 goat anti-rabbit antibody (Thermo Fisher Scientific A11034) in blocking solution, and washed in PBSTx. All fluorescently labeled parasites were counterstained with DAPI (1 µg/ml), cleared in 80% glycerol, and mounted on slides with Vectashield (Vector Laboratories).

Confocal imaging of fluorescently labeled samples was performed on either a Zeiss LSM700 or a Nikon A1 Laser Scanning Confocal Microscope. Unless otherwise stated all fluorescence images are taken at the anatomical level of the tegumental cell bodies (see *Figure 1D* for approximate location) and represent maximum intensity projections. To perform cell counts, cells were manually counted in maximum intensity projections derived from confocal stacks. We used two types of measurements to normalize cell counts between samples. In cases where we determined the number of cells in a particular region of the parasite (e.g., tegument) we collected confocal stacks and normalized the number of cells by total volume of the stack in $\mu m^3$. In cases where we determined the total number of labeled foci throughout the entire depth of the parasite (e.g. EdU counts), we collected confocal stacks and normalized the number of cells to the length of the parasite in the imaged region in mm. Brightfield images were acquired on a Zeiss AxioZoom V16 equipped with a transmitted light base and a Zeiss AxioCam 105 Color camera.

## Fluorescence activated cell sorting

Freshly perfused worms were either exposed to 100 Gy of Gamma Irradiation on a J.L. Shepard Mark I-30 Cs$^{137}$ source or left alone to serve as controls, then cultured for one week. After one week, males were separated from female worms by incubation in a 0.25% solution of tricaine (*Collins et al., 2013*). Male worms were amputated to remove the head and testes, and the bodies of the worms were collected. These worm bodies were briefly minced with a razor blade and then suspended in a 0.5% solution of Trypsin/EDTA (Sigma T4174) in PBS. The worms were then triturated for approximately 15 min until the solution became turbid and no large pieces of worms were left. The cells were then centrifuged at 500 *g* for 10 m at 4°C. Next the cells were resuspended in 1 ml of Basch media with 10 µl of RQ1 DNAse (Promega M6101) and incubated for 10 min at RT. The cells were centrifuged again at 500 *g* for 10 min at 4°C. The cells were resuspended in staining media (0.5% BSA, 2 mM EDTA in PBS) and incubated in anti-TSP-2 polyclonal antibody (1:1000 dilution) for 45 min in the dark at 4°C. The cells were centrifuged again at 500 *g* for 10 min at 4°C. The cells were then resuspended in staining media and incubated in Hoechst 33342 (18 µg/ml) (Sigma B2261) and goat anti-rabbit AF488 (Thermo Fisher Scientific A11034) (1:1000 dilution) for 1 hr at RT in the dark. The cells were centrifuged once again at 500 *g* for 10 min at 4°C. The cells were then resuspended in staining media containing Hoechst 33342 (18 µg/ml) and propidium iodide (1 µg/ml) (Sigma-Aldrich P4170) and then filtered through a 40 µm cell strainer. Filtered cells were then sorted on a FACSAria Fusion (BD Biosystems) with a 100 µm nozzle either into staining media for confocal imaging or directly into Trizol LS (Thermo Fisher Scientific 10296–010) for RNAseq experiments. For all FACS experiments, a Hoechst threshold was applied to exclude debris and improve the efficiency of sorting.

## Transcriptional profiling by RNA sequencing

RNA was extracted from purified cells (>40000 'Neoblast', >4000 'TSP-2$^+$', and 80000 'IR Rest' cells per biological replicate) collected from three independent FACS runs using Trizol LS (Thermo Fisher Scientific 10296–010). Libraries for RNAseq analysis were generated using the SMART-seq2 kit (Clontech) and reads obtained by Illumina sequencing. The total number of reads per gene was determined by mapping the reads to the *S. mansoni* genome using STAR (version 020201) (*Dobin et al., 2013*). *S. mansoni* genome sequence and GTF files used for mapping were acquired from Wormbase Parasite (*Howe et al., 2017*). Pairwise comparisons of differential gene expression were performed with DESeq2 (version 1.12.2) (*Love et al., 2014*). To determine which genes showed the highest level of enrichment in the various cell populations we also performed Model Based clustering using the MBCluster.seq package in R (*Si et al., 2014*). This clustering analysis was only performed on genes that had more than 200 total reads from the Neoblast, TSP-2$^+$, and IR-REST cell populations. Raw data for RNAseq of FACS sorted cells are available at NCBI under the accession numbers as follows: Neoblasts (ERS1987942, ERS1987945, ERS1987957), TSP-2 HI (ERS1987946, ERS1987958, ERS1987961) and IR Rest (ERS1987948, ERS1987962, ERS1987958). For RNAseq analysis of *zfp-1-1(RNAi)* parasites, Illumina reads for three biological replicates were mapped to the schistosome genome using STAR and differential gene expression changes were measured using DESeq2. The statistical enrichment of the various clusters of genes that were down-regulated following *zfp-1-1(RNAi)* ($\log_2 < -0.5$, padj $< 0.05$) was measured using a Fisher's exact test in R. Data used for the analysis is provided in *Supplementary file 6*. RNAseq datasets for the *zfp-1-1*(RNAi) experiments are available at NCBI through the accession number GSE106693.

## Western blotting to detect TSP-2

To generate protein lysates, RNAi treated male parasites were separated with 0.5% tricaine, their heads and testes were amputated, the remaining somatic tissue was homogenized in 100 µl of sample buffer (236 mM Tris pH 6.7, 128 mM $H_3PO_4$, 4% SDS, 20% Glycerol, 10 mM DTT, and protease inhibitors (Roche cOmplete, Mini, EDTA-free)). Homogenized samples were incubated at 42°C for 45 min and alkylated with N-ethylmaleimide for 40 min at 37°C. Protein concentrations were determined by BCA assays, 40 µg of lysate was separated by SDS PAGE, proteins were transferred to nitrocellulose membranes, membranes were blocked in Li-Cor Odyssey Blocking Buffer, incubated in rabbit anti-TSP-2 (1:5000) and mouse anti-Actin (0.25 µg/ml, Monoclonal JLA20, Developmental Studies Hybridoma Bank), washed in TBST, and incubated in secondary antibodies (1:10,000 goat anti-rabbit IRDye 680 RD, 1:15,000 goat anti-mouse IgM IRDye 800CW, Li-Cor). Blots were imaged using a Li-Cor Odyssey Infrared Imager.

## Protein alignments and phylogenetic tree

To estimate the evolutionary relationship between the various flatworm ZFP-1 family members, protein sequences of these family members were aligned using Guidance (http://guidance.tau.ac.il) (settings: MSA Algorithm = MAFFT; –maxiterate 1000 –genafpair; number of alternative guide-trees: 100). Columns in the sequence alignment with a confidence score below 0.050 were removed and a tree was generated using RAxML (version 8.0.0) (options –T 4 –f a –p 11111 –x 1111 -# 1000 –m PROTGAMMAAUTOF). Sequences used for phylogenetic analysis were recovered from Wormbase Parasite (*Howe et al., 2017*) (https://parasite.wormbase.org), Planmine (*Brandl et al., 2016*)(http://planmine.mpi-cbg.de), the *Gyrodactylus salaris* genome database (http://invitro.titan.uio.no/gyrodactylus/index.html) (*Hahn et al., 2014*), and the *Macrostomum lignano* genome initiative database (*Simanov et al., 2012*) (http://www.macgenome.org). A FASTA formatted file can be found in *Supplementary file 7*. *S. mansoni* ZFP-1 and ZFP-1–1 DNA binding motifs were predicted using the ZFModels web server at http://stormo.wustl.edu/ZFModels/ (*Gupta et al., 2014*).

## Acknowledgements

We thank Megan McConathy for comments on the manuscript. Mice and *B. glabrata* snails were provided by the NIAID Schistosomiasis Resource Center of the Biomedical Research Institute (Rockville, MD) through NIH-NIAID Contract HHSN272201000005I for distribution through BEI Resources. NVG

is an investigator of the Howard Hughes Medical Institute. The authors declare no competing interests.

## Additional information

### Funding

| Funder | Grant reference number | Author |
|---|---|---|
| National Institutes of Health | R01AI121037 | James J Collins III |
| Wellcome | 107475/Z/15/Z | Matthew Berriman<br>James J Collins III |
| National Institutes of Health | R01GM094575 | Nick V Grishin |
| Welch Foundation | I1505 | Nick V Grishin |
| National Health and Medical Research Council | | Alex Loukas |

The funders had no role in study design, data collection and interpretation, or the decision to submit the work for publication.

### Author contributions

George R Wendt, Conceptualization, Formal analysis, Investigation, Writing—original draft; Julie NR Collins, Investigation, Writing—review and editing; Jimin Pei, Formal analysis, Investigation; Mark S Pearson, Resources, Writing—review and editing; Hayley M Bennett, Formal analysis, Writing—review and editing; Alex Loukas, Matthew Berriman, Supervision, Funding acquisition, Writing—review and editing; Nick V Grishin, Supervision, Funding acquisition; James J Collins III, Conceptualization, Formal analysis, Supervision, Funding acquisition, Writing—original draft, Project administration

### Author ORCIDs

Mark S Pearson (iD) http://orcid.org/0000-0002-0002-1544
Matthew Berriman (iD) http://orcid.org/0000-0002-9581-0377
James J Collins III (iD) https://orcid.org/0000-0001-5237-1004

### Ethics

Animal experimentation: In adherence to the Animal Welfare Act and the Public Health Service Policy on Humane Care and Use of Laboratory Animals, all experiments with and care of vertebrate animals were performed in accordance with protocols approved by the Institutional Animal Care and Use Committee (IACUC) of the UT Southwestern Medical Center (protocol approval number APN 2014-0072).

### Decision letter and Author response

Decision letter https://doi.org/10.7554/eLife.33221.032
Author response https://doi.org/10.7554/eLife.33221.033

## Additional files

### Supplementary files

• Supplementary file 1. Pairwise comparisons of transcriptional profiles of neoblasts, TSP-2+ cells, and IR Rest cells.
DOI: https://doi.org/10.7554/eLife.33221.019

• Supplementary file 2. Results of model-based clustering analysis to define genes whose expression is enriched in either neoblasts, TSP-2+ cells, or IR Rest cells.
DOI: https://doi.org/10.7554/eLife.33221.020

• Supplementary file 3. Table of candidate tegument-associated genes, their abbreviations, their gene expression cluster, and their expression pattern.
DOI: https://doi.org/10.7554/eLife.33221.021

• Supplementary file 4. Pairwise comparisons of transcriptional profiles of control(RNAi) parasites versus *zfp-1-1*(RNAi) parasites.
DOI: https://doi.org/10.7554/eLife.33221.022

• Supplementary file 5. Table of gene names, abbreviations, and oligonucleotides sequences from this study.
DOI: https://doi.org/10.7554/eLife.33221.023

• Supplementary file 6. Table of results from Fisher's exact test to define how *zfp-1–1* RNAi treatment affects genes expressed in various gene expression clusters.
DOI: https://doi.org/10.7554/eLife.33221.024

• Supplementary file 7. FASTA formatted file with various flatworm ZFP-1 family protein sequences that were used for generating protein alignments and phylogenetic trees.
DOI: https://doi.org/10.7554/eLife.33221.025

• Transparent reporting form
DOI: https://doi.org/10.7554/eLife.33221.026

## Major datasets

The following datasets were generated:

| Author(s) | Year | Dataset title | Dataset URL | Database, license, and accessibility information |
|---|---|---|---|---|
| Wellcome Sanger Institute | 2017 | Characterising_Schistosoma_mansoni_stem_cell_populations | https://www.ncbi.nlm.nih.gov/bioproject/416469 | Publicly available at NCBI BioProject (accession numbers: ERS1987962, ERS1987961, ERS1987958, ERS1987958, ERS1987957, ERS1987948, ERS1987946, ERS1987945, ERS1987942) |
| Collins J, Wendt G | 2017 | Transcriptional Profiling of Schistosoma mansoni zfp-1-1(RNAi) parasites | https://www.ncbi.nlm.nih.gov/geo/query/acc.cgi?acc=GSE106693 | Publicly available at the NCBI Gene Expression Omnibus (accession number: GSE106693) |

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
