## [Decision Letter]

Thank you for submitting your article "Flatworm-specific transcriptional regulators promote the specification of tegumental progenitors in *Schistosoma mansoni*" for consideration by *eLife*. Your article has been reviewed by two peer reviewers, and the evaluation has been overseen by a Reviewing Editor and Marianne Bronner as the Senior Editor. The following individuals involved in review of your submission have agreed to reveal their identity: Peter W Reddien (Reviewer #2); Carolyn E Adler (Reviewer #3). We apologize for the excessive delay in rendering this opinion but hope you will find the comments constructive and supportive of your interesting work.

The reviewers have discussed the reviews with one another and the Reviewing Editor has drafted this decision to help you prepare a revised submission.

Summary:

Schistosome parasites pose a major health hazard and they evolved fascinating mechanisms to be able to survive within their hosts. As part of this survival strategy, they maintain a tegument (outer protective epithelial layer) that aids in their survival. The authors have developed a method to label the tegument and then used this to understand the mechanisms by which the tegument is maintained by stem cell growth. They go on to identify a number of Zn-finger transcription factors that play a key role in the maintenance of this epithelium through the specification of the progenitor cells for the tegument. The work here marks a significant advance in our understanding of the biology of these fascinating animals and has important implications in the control of their life cycle.

Essential revisions:

1) Subsection “FACs purification and molecular characterization of neoblasts and TSP-2+ cells”: Please show the data for the imaging of cells with intermediate levels of TSP in the FACs prep. It is stated "Visual examination of these cells found that they did not possess high levels of TSP-2 surface labeling. Instead, these "TSP-2 Intermediate" cells had either no TSP-2 surface labeling or had pieces of TSP-2-labeled debris attached to their surface." It is important to show representative images of this.

2) Figure 1K and Figure 1—figure supplement 1: Please report what fraction of tegument cells are tsp-2+? What fraction are sm13+? Relatedly, what fraction of tsp-2+ cells are sm13+, and what fraction of zfp1-1 cells are sm13+?

3) Figure 2G: Some kind of graphical summary of the findings is needed. There are deep tsp-2+ cells; superficial tsp-2+ cells; rare tsp-2+/tegument+ cells; and tsp-2-; tegument+ cells. From the in-situ survey of cluster-enriched genes, you have found genes for which expression spanned these stages – helping make the lineage argument. However, the reader has to work a bit hard to tease this information out of the images in Figure 2G, the text, and Supplementary data. A cartoon summary of the cell stages and which genes were found to be expressed in multiple categories should be provided.

4) In Figure 3, the claim that zfp-1 and zfp1-1 are exclusively expressed in neoblasts, and that they are not present in the same cells, should be supported by some quantification. Also identify what part of the animal is shown in Figure 3M. Some anatomical context should be provided for non-schistosome biologists.

5) In the Introduction, the authors state that understanding the development of the syncytial tegument is critical for bolstering knowledge of these parasites for drug development, but this aspect of tegument biology is never mentioned again. Many questions remain about the biology of this tissue which are relevant to data presented in this paper. After zfp-1-1 knockdown, does nuclear spacing increase? If not, is animal viability (long and short-term) affected (as in free-living flatworms after inhibiting formation of the epidermis)? While details of this result might be better suited for future work, the authors should at least make some mention of the potential value of zfp1-1 RNAi for such studies and any possible preliminary thoughts.

---

## [Author Response]

Essential revisions:1) Subsection “FACs purification and molecular characterization of neoblasts and TSP-2+ cells”: Please show the data for the imaging of cells with intermediate levels of TSP in the FACs prep. It is stated "Visual examination of these cells found that they did not possess high levels of TSP-2 surface labeling. Instead, these "TSP-2 Intermediate" cells had either no TSP-2 surface labeling or had pieces of TSP-2-labeled debris attached to their surface." It is important to show representative images of this.

We agree with the reviewers. We have added images of the sorted “TSP-2 Intermediate” cells in Figure 2—figure supplement 1. Arrows within the inset indicate cells with no TSP-2 labeling, cells with only partial TSP-2 labeling, and acellular debris.

2) Figure 1K and Figure 1—figure supplement 1: Please report what fraction of tegument cells are tsp-2+? What fraction are sm13+? Relatedly, what fraction of tsp-2+ cells are sm13+, and what fraction of zfp1-1 cells are sm13+?

Over 3 experiments we observed a total of 5 *tsp-2*^+^ tegument cells out of 3074 total tegument cells counted (~0.2%) and we have added this data to the manuscript in subsection “Definitive tegumental cells express *calpain, npp-5, annexin* and *gtp-4* but not *tsp-2”*. One issue with this analysis is the low expression level of *tsp-2* observed in tegument cells (such as in Figure 1K). In order to visualize the low levels of *tsp-2* in these cells, the gain on the confocal microscope had to be turned up to the point of oversaturating what we elsewhere describe as *tsp-2*^+^ cells. The aforementioned experiments where we quantified *tsp-2* expression in tegument cells were performed with a significantly lower gain in the *tsp-2* channel. This could result in our inability to detect rare *tsp-2*^low^ tegument cells. As such, we are only confident in saying that *tsp-2*^+^ tegument cells are extremely rare (less than 1% of all tegument cells), and when co-expression is observed, *tsp-2* is expressed at very low levels.

With respect to questions regarding the expression of *sm13* in tegument cells, we apologize for the lack of clarity in the initial submission of the manuscript. Figure 1—figure supplement 1 H-L shows dextran/FISH experiments and double FISH experiments demonstrating that there are virtually no *sm13*^+^ tegument cells. We have added counts of this data to subsection “Definitive tegumental cells express *calpain, npp-5, annexin* and *gtp-4* but not *tsp-2”*, indicating that we observed only 1 sm13^+^ tegmental cell in 1826 observed.

With respect to questions regarding co-expression of *sm13* with *tsp-2*, we apologize for the lack of clarity in the manuscript. Figure 1—figure supplement 1G shows a Venn diagram in the inset indicating the relative overlap of *tsp-2* and *sm13*. All *sm13* cells are *tsp-2*^+^. Though the majority of *tsp-2*^+^ cells are *sm13*^+^, some *tsp-2* cells are *sm13*^-^. This appears to relate to the position of the cells along the dorsal-ventral axis. Virtually all dorsally located *tsp-2*^+^ cells are *sm13*^+^ whereas almost no parenchymal *tsp-2*^+^ cells are *sm13*^+^ (See Figure 2G). The images shown in Figure 1—figure supplement 1 are all taken at the level of the tegument, hence the high overlap of *tsp-2* and *sm13*. We now indicate the anatomical location of the imaging in the figure legend for Figure 1—figure supplement 1G.

With respect to questions regarding co-expression of *sm13* with *zfp-1-1*, there is almost no overlap between these two populations (1 double positive cell out 248 *sm13^+^* cells examined). This result is reflected in a new supplemental figure: Figure 3—figure supplement 3. As mentioned previously, *sm13* is expressed almost exclusively in superficial *tsp-2*^+^ cells, whereas *zfp-1-1* is expressed exclusively in more parenchymal *tsp-2*^+^ cells (Figure 3M). We now discuss in the manuscript in subsection “An RNAi screen identifies *zfp-1* and *zfp-1-1* as potential regulators of tegument development**”**.

3). Figure 2G: Some kind of graphical summary of the findings is needed. There are deep tsp-2+ cells; superficial tsp-2+ cells; rare tsp-2+/tegument+ cells; and tsp-2-; tegument+ cells. From the in-situ survey of cluster-enriched genes, you have found genes for which expression spanned these stages – helping make the lineage argument. However, the reader has to work a bit hard to tease this information out of the images in Figure 2G, the text, and Supplementary data. A cartoon summary of the cell stages and which genes were found to be expressed in multiple categories should be provided.

We agree that a graphical summary will help make these data easier to interpret. Such as summary has now been included in the manuscript as Figure 2—figure supplement 3.

4) In Figure 3, the claim that zfp-1 and zfp1-1 are exclusively expressed in neoblasts, and that they are not present in the same cells, should be supported by some quantification. Also identify what part of the animal is shown in Figure 3M. Some anatomical context should be provided for non-schistosome biologists.

We apologize for the lack of clarity in the manuscript. Figure 3J shows that zfp-1 is expressed in nanos2^+^ neoblasts but not in tsp-2^+^ cells whereas zfp-1-1 is expressed in tsp-2^+^ cells but not in nanos2^+^ neoblasts. A direct comparison of the two cell populations also reveals no overlap between the populations as detectable by FISH (See subsection “*zfp-1* and *zfp-1-1* are members of a family of flatworm-specific DNA binding proteins whose homolog in planarians regulates epidermal lineage specification”). Additionally, we have added an arrow in Figure 3M that clarifies the dorsal-ventral axis. To further orient readers, we have added the following sentence to the Materials and methods section describing the anatomical region images were acquired from: “Unless otherwise stated all fluorescence images are taken at the anatomical level of the tegumental cell bodies (see Figure 1D for approximate location)”

5) In the Introduction, the authors state that understanding the development of the syncytial tegument is critical for bolstering knowledge of these parasites for drug development, but this aspect of tegument biology is never mentioned again. Many questions remain about the biology of this tissue which are relevant to data presented in this paper. After zfp-1-1 knockdown, does nuclear spacing increase? If not, is animal viability (long and short-term) affected (as in free-living flatworms after inhibiting formation of the epidermis)? While details of this result might be better suited for future work, the authors should at least make some mention of the potential value of zfp1-1 RNAi for such studies and any possible preliminary thoughts.

We are very interested in the consequences of disrupting tegument biogenesis and are working to develop the tools that will allow us to study this question in a physiologically relevant context (i.e., in the presence of the host immune system). In Figure 5B-C we show that knocking down *zfp-1* or *zfp-1-1* results in a ~10% and ~30% reduction in tegument cell body density, respectively. In addition to the decrease in the total number of tegumental cells, we also observed an inability of *zfp-1-1*(RNAi) animals, but not *control(RNAi)* or *zfp-1*(RNAi) animals, to remain attached to the plate during the course of our RNAi experiments. We have included these data in the revised manuscript (Figure 5F). Plate attachment is mediated by the parasite’s ventral sucker and since these organs are covered by tegument, tegumental maintenance is potentially required for plate attachment. It is also possible the plate attachment phenotype reflects a secondary effect of reduced tegumental maintenance since plate attachment also certainly relies a properly functioning neuromuscular system. We believe this supports the model that *zfp-1-1*, and perhaps tegumental maintenance are important for the parasite. We have added a section to discuss these results (Discussion section). In addition, we have updated to the final paragraph of the discussion to include our thinking about how our understanding of *zfp1-1* could be used in the future to combat these worms. In particular, we say:

*“*Furthermore, since the tegument is critical to parasite biology, understanding the tegument lineage, and the molecular targets of *zfp-1* homologues, in diverse flatworms could suggest novel therapies to blunt tegument development in this important group of parasites*”.*